# Global kinome profiling reveals DYRK1A as critical activator of the human mitochondrial import machinery

Corvin Walter[1,2,12], Adinarayana Marada[1,12], Tamara Suhm[1], Ralf Ernsberger[1], Vera Muders[1], Cansu Kücükköse[1,2], Pablo Sánchez-Martín[1], Zehan Hu [3], Abhishek Aich [4,5], Stefan Loroch[6], Fiorella Andrea Solari[6], Daniel Poveda-Huertes [1], Alexandra Schwierzok[1], Henrike Pommerening[1], Stanka Matic[1], Jan Brix[1], Albert Sickmann [6], Claudine Kraft [1,7], Jörn Dengjel [3], Sven Dennerlein[4], Tilman Brummer [8,9,10], F.-Nora Vögtle[1,7,11✉] & Chris Meisinger [1,7,9✉]

The translocase of the outer mitochondrial membrane TOM constitutes the organellar entry gate for nearly all precursor proteins synthesized on cytosolic ribosomes. Thus, TOM presents the ideal target to adjust the mitochondrial proteome upon changing cellular demands. Here, we identify that the import receptor TOM70 is targeted by the kinase DYRK1A and that this modification plays a critical role in the activation of the carrier import pathway. Phosphorylation of TOM70[Ser91] by DYRK1A stimulates interaction of TOM70 with the core TOM translocase. This enables transfer of receptor-bound precursors to the translocation pore and initiates their import. Consequently, loss of TOM70[Ser91] phosphorylation results in a strong decrease in import capacity of metabolite carriers. Inhibition of DYRK1A impairs mitochondrial structure and function and elicits a protective transcriptional response to maintain a functional import machinery. The DYRK1A-TOM70 axis will enable insights into disease mechanisms caused by dysfunctional *DYRK1A*, including autism spectrum disorder, microcephaly and Down syndrome.

[1] Institute of Biochemistry and Molecular Biology, ZBMZ, Faculty of Medicine, University of Freiburg, Freiburg, Germany. [2] Faculty of Biology, University of Freiburg, Freiburg, Germany. [3] Department of Biology, University of Fribourg, Fribourg, Switzerland. [4] Department of Cellular Biochemistry, University Medical Center Göttingen, Göttingen, Germany. [5] Cluster of Excellence "Multiscale Bioimaging: from Molecular Machines to Networks of Excitable Cells" (MBExC), University of Göttingen, Göttingen, Germany. [6] Leibniz Institut für Analytische Wissenschaften - ISAS - e.V., Dortmund, Germany. [7] CIBSS - Centre for Integrative Biological Signalling Studies, University of Freiburg, Freiburg, Germany. [8] Institute of Molecular Medicine, ZBMZ, Faculty of Medicine, University of Freiburg, Freiburg, Germany. [9] BIOSS Centre for Biological Signalling Studies, University of Freiburg, Freiburg, Germany. [10] German Cancer Consortium DKTK Partner Site Freiburg, German Cancer Research Center (DKFZ), Heidelberg, Germany. [11] Present address: Center for Molecular Biology of Heidelberg University (ZMBH), Heidelberg, Germany. [12] These authors contributed equally: Corvin Walter, Adinarayana Marada. ✉email: n.voegtle@zmbh.uni-heidelberg.de; chris.meisinger@biochemie.uni-freiburg.de

Mitochondria are essential organelles in eukaryotic cells and play numerous functions important e.g. for energy supply, biosynthesis of metabolites, and cofactors or regulation of programmed cell death. All these functions rely on an extensive and functional organellar proteome, which is mainly built by import of nuclear-encoded precursor proteins from the cytosol[1–6]. Dedicated signal sequences direct the cytosolic precursors to the translocase of the outer membrane (TOM), from where they are sorted to their suborganellar destinations[7–13]. The two most comprehensive subproteomes of mitochondria[14], the inner membrane and the matrix are built via two main import routes, termed the carrier- and presequence-import pathway. Metabolite carriers are polytopic inner membrane proteins and possess internal, non-cleavable targeting signals. They are typically delivered by cytosolic chaperones to the import receptor Tom70 and then translocated via TOM into the intermembrane space[15–21]. The TIM22 translocase then mediates sorting and assembly in the inner membrane[22–25]. Presequences are N-terminal extensions that serve as targeting signals for import into the matrix and are mainly recognized and bound by the import receptors Tom20 and Tom22. After translocation across the outer membrane the presequence precursors are handed over to the TIM23 complex from where they are imported into the matrix or, in case of single spanning membrane proteins laterally sorted into the inner membrane[14,26–28].

Recent studies revealed that the main import receptors Tom20 and Tom70 are rather loosely associated with the core TOM complex, which consists of Tom40, Tom22, and the small Tom proteins Tom5, 6, and 7 [19,21,29–32]. While the mitochondrial import translocases have been considered for long time static and constitutively active entities, studies in the model organism *S. cerevisiae* have shown that components of the organellar protein entry gate TOM are modified by cytosolic protein kinases and allow regulation of the import flux upon changing cellular conditions[33–35]. However, if such regulatory mechanisms exist for the human mitochondrial import machinery remains enigmatic. The human kinome comprises 550 different kinases and thus exceeds the yeast kinome by more than 5-fold[36]. This large number of kinases makes the discovery of signaling pathways that regulate mitochondrial protein biogenesis in humans challenging.

Here, we performed a global screen for kinase candidates of the mitochondrial import receptor TOM70 and identified DYRK1A as critical activator of the human import machinery. Interestingly, dysfunctional *DYRK1A* was described to cause an array of disease phenotypes including intellectual disability, speech delay, motor deficits, autism, and microcephaly (*DYRK1A-related syndrome*). Moreover, *DYRK1A* is localized in the Down syndrome critical region of chromosome 21 and is considered a strong candidate for learning defects associated with Down syndrome[37–42]. Several of these clinical manifestations of *DYRK1A*-related syndrome, particularly in their combinations, suspiciously resemble phenotypes described of mitochondrial diseases[18,43–45]. This led us to speculate that dysfunctional DYRK1A signaling affects mitochondrial integrity and that this is mediated by impaired protein import.

We found that the import receptor TOM70 is almost quantitatively phosphorylated in human cells in vivo and that DYRK1A exclusively targets residue Serine 91. Furthermore, phosphorylation of TOM70 is crucial for activation of the metabolite carrier import pathway. We analyzed the impact of $TOM70^{Ser91}$ phosphorylation on mitochondrial protein biogenesis on the molecular level and found that DYRK1A phosphorylation at $TOM70^{Ser91}$ does not affect binding of precursor proteins, but strongly stimulates the interaction of the import receptor with the core TOM machinery. Consequently, selective inhibition of DYRK1A signaling in vivo leads to impairment of crucial mitochondrial functions, which is mitigated by a protective transcriptional response and remodeling of the import machinery. Our findings reveal an unexpected link of mitochondrial protein biogenesis and DYRK1A signaling and can enable an understanding of the pathophysiological consequences of dysfunctional *DYRK1A* in *DYRK1A*-related syndrome, Autism Spectrum Disorder, and Down syndrome.

## Results

**DYRK1A targets the mitochondrial import receptor TOM70.** To search for kinase candidates targeting the human mitochondrial import machinery we employed a *KinaseFinder* platform which is based on a radiometric filter binding assay[46] using $^{33}$P-ATP (ProQinase (Freiburg, Germany)) and allows to test for 339 different human protein kinases experimentally in an unbiased approach. The soluble cytosolic domain of human TOM70 ($TOM70_{cd}$), which represents the main import receptor for the carrier import pathway, was used as substrate (Supplementary Fig. 1a). The two DYRK1 isoforms DYRK1A and DYRK1B showed the by far highest scores in phosphorylating $TOM70_{cd}$ (Fig. 1a). Furthermore, the kinase activities of both, DYRK1A and DYRK1B, were dependent on the amount of added TOM70 protein (Supplementary Fig. 1b).

To validate these results further we incubated $TOM70_{cd}$ in the presence or absence of DYRK1A and monitored phosphorylation activity via phosphate affinity (Phos-tag) SDS-PAGE that retards the gel mobility of phosphorylated proteins[33,34,47,48]. $TOM70_{cd}$ migrated slower upon incubation with DYRK1A (Fig. 1b, lanes 5 and 6), indicating a DYRK1A-dependent phosphorylation of $TOM70_{cd}$. In contrast, the receptor domains of $TOM22_{cd}$ and $TOM20_{cd}$ did not show a changed migration upon incubation with DYRK1A indicating that DYRK1A and DYRK1B exclusively target the TOM70 receptor (Fig. 1b, lanes 1–4 and Supplementary Fig. 1a).

Notably, DYRK1A and DYRK1B have not been linked to mitochondria before. DYRK1A and DYRK1B are two isoforms with almost identical catalytic sites[49]. DYRK1A is ubiquitously expressed, located in the nucleus and cytosol and involved in many different cellular processes including cell differentiation, proliferation, and survival[49–52]. The isoform DYRK1B has similar functions as DYRK1A, but is only expressed in a very limited spectrum of cell types[49]. Thus, we focused on a possible link between DYRK1A and mitochondria and assessed its potential role in regulating the human mitochondrial protein import machinery.

To identify which residues of TOM70 are targeted by DYRK1A, we profiled phosphorylation sites in isolated mitochondria from HEK293T cells by phosphoproteomics. Phosphopeptide enrichment using ERLIC followed by LC-MS/MS analysis revealed three phosphorylated residues on TOM70: Thr85, Ser91, and Ser110 (Supplementary Fig. 1c and Methods section). To test which of these residues might represent targets of DYRK1A, we purified variants of the TOM70 receptor domain harboring alanine residues that cannot be phosphorylated at the respective positions (Supplementary Fig. 1d). In vitro phosphorylation of these domains by DYRK1A followed by Phos-tag gel electrophoresis revealed that the wild-type (WT) TOM70, $TOM70^{T85A}$, and $TOM70^{S110A}$ but not the $TOM70^{S91A}$ domains were phosphorylated by DYRK1A (Fig. 1c). This indicates that DYRK1A can phosphorylate human TOM70 at residue Ser91. While these in vitro assays revealed Ser91 as target of DYRK1A we wondered about the abundance and relevance of this phosphorylation site in vivo. For this, we analyzed mitochondria isolated from HEK293T cells by two approaches. Firstly, Phos-tag gel electrophoresis of non-treated mitochondria revealed a

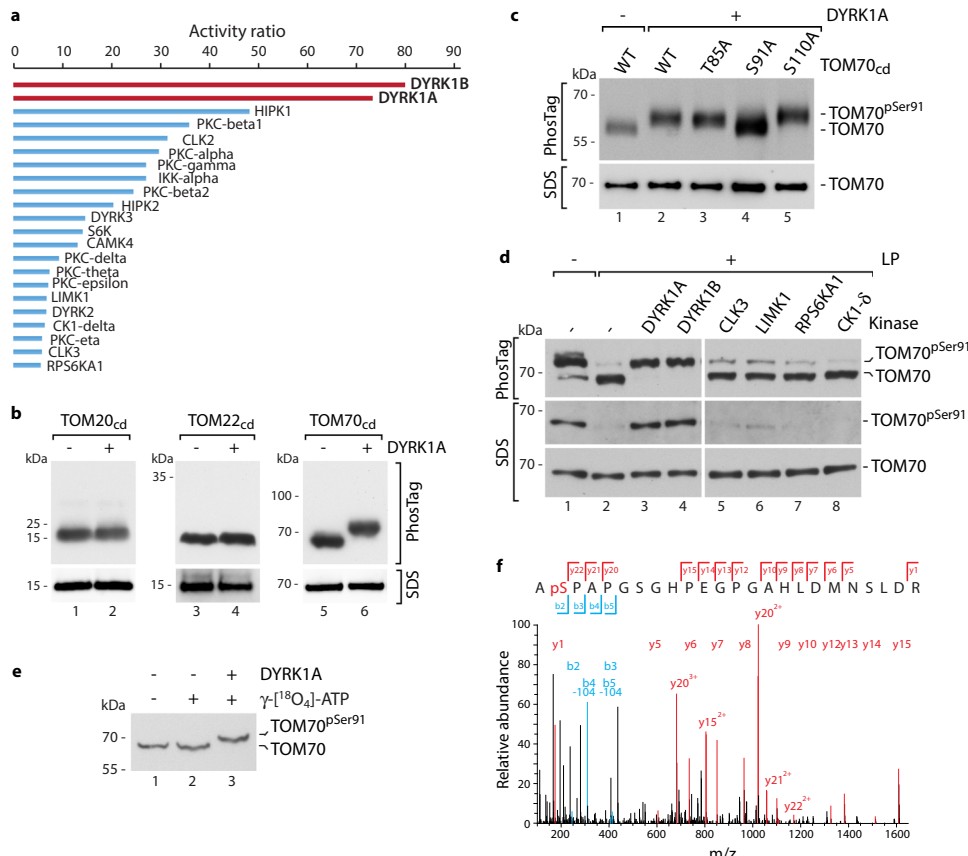

**Fig. 1 Identification of DYRK1A as kinase of the mitochondrial import receptor TOM70. a** Results of *KinaseFinder assay* (ProQinase™) for human TOM70$_{cd}$ tested for 339 different protein kinases. Shown are activity ratios of $^{33}$P-ATP-radiometric filter binding assays. **b** In vitro phosphorylation assay of soluble cytosolic domains (cd) of the human import receptors TOM70, TOM20, and TOM22. Proteins were incubated in the presence or absence of human DYRK1A. Samples were analyzed via Phos-tag electrophoresis, SDS-PAGE and immunoblotting using the respective TOM antibody (Supplementary Table 1). **c** In vitro kinase assay of indicated TOM70$_{cd}$ variants in the absence or presence of DYRK1A. Samples were analyzed by Phos-tag gel electrophoresis (upper panel) or standard SDS-PAGE (lower panel) followed by immunoblotting using TOM70 antisera. **d** Analysis of isolated mitochondria from HEK293T cells incubated in the presence or absence of lambda phosphatase (LP). Where indicated samples were incubated with the respective kinase after LP treatment. Samples were analyzed by Phos-tag gels (upper panel) or standard SDS gels (middle and lower panel). Upper and lower panels show immunoblots stained with TOM70 antibody and middle panel was stained with TOM70$^{Ser91}$ phosphospecific antibody. **e** Isolated mitochondria were incubated with γ-[$^{18}O_4$]-ATP and DYRK1A where indicated. Samples were analyzed via Phos-tag gels and immunoblotting using TOM70 antibody. **f** MS/MS spectrum from mitochondria incubated with γ-[$^{18}O_4$]-ATP shown in **e** localizing the DYRK1A-dependent phosphorylation of TOM70 to Ser91. Identified b- and y-ions are listed in blue and red, respectively. The b2-y22 ion pair localizes the phosphorylation unambiguously to Ser91. The amino acid sequence of the respective tryptic peptide TOM70$^{90-113}$ is indicated. Unprocessed immunoblots are reported in the Source Data File.

prominent TOM70 immunoreactive band that was accessible to treatment with lambda phosphatase (LP) (Fig. 1d, upper panel, lanes 1 and 2). This indicates that TOM70 is almost quantitatively phosphorylated in vivo. Secondly, analysis of the same samples via SDS-PAGE and immunodecoration with phospho-Ser91 specific TOM70 antiserum revealed loss of the signal upon LP treatment while the general TOM70 immunoreactivity was not affected (Fig. 1d, middle and lower panel, lanes 1 and 2 and Supplementary Fig. 1e). Notably, in both approaches, addition of DYRK1A or DYRK1B kinase after phosphatase (LP) treatment resulted in redetection of the Ser91 phosphorylation (Fig. 1d, lanes 3 and 4) while kinases with low scores in our kinase screen (Fig. 1a) revealed no significant changes (Fig. 1d, lanes 5–8 and Supplementary Fig. 1f, g). Interestingly, HIPK1 and CLK2, members of the DYRK kinase family and the closely related CLK family[51], respectively, which scored in a medium-range in the *Kinase Finder* screen, were also able to phosphorylate TOM70 Ser91, however, to a lesser extent compared to DYRK1A and DYRK1B (Supplementary Fig. 1h). To analyze if further subunits of the TOM import machinery are targeted by DYRK1A we

incubated isolated and phosphatase (LP) treated mitochondria in the presence or absence of DYRK1A and included γ-[$^{18}O_4$]-ATP to detect phosphorylated residues directly dependent on the presence of DYRK1A by phosphoproteomics (Fig. 1e). LC-MS/MS analysis revealed residue Ser91 of TOM70 as the only site of the entire import machinery that was specifically phosphorylated by DYRK1A (Fig. 1f and Methods section).

In summary, our data show that human TOM70 is nearly quantitatively phosphorylated at Ser91 in vivo and that this position is specifically targeted by DYRK1A in vitro and in isolated mitochondria.

**DYRK1A signaling is required for functional mitochondria in vivo.** To investigate if DYRK1A also targets TOM70 in vivo and if its phosphorylation of TOM70 impacts on mitochondrial protein biogenesis (e.g. displaying an activatory or inhibitory role) we treated cells with INDY, a benzothiazole derivative that was developed as potent and selective DYRK1A inhibitor[53] and can indeed impair TOM70 Ser91 phosphorylation by DYRK1A

in vitro (Supplementary Fig. 2a). Phos-tag gel electrophoresis of cells incubated overnight in the presence or absence of INDY revealed a significant change in the ratio of phosphorylated to non-phosphorylated TOM70 indicating that DYRK1A targets TOM70 in vivo (Supplementary Fig. 2b). Next, we aimed to analyze how mitochondrial functions would be affected upon compromised DYRK1A activity. At first, respiratory activity in the presence or absence of INDY was tested by real-time respirometry and revealed a significantly impaired function of the OXPHOS machinery upon DYRK1A inhibition (Fig. 2a). We also found a strong decrease in cell growth upon DYRK1A inhibition (Supplementary Fig. 2c). When we analyzed mitochondrial morphology by fluorescence microscopy, we observed a severe loss of the typical elongated, tubular mitochondrial structure in INDY-treated cells (Fig. 2b, c). Moreover, the INDY-induced aberrant mitochondrial morphology seemed to be in part mediated by the TOM70 Ser91 phosphorylation as overexpression of TOM70$^{WT}$ but not of the non-phosphorylatable TOM70$^{S91A}$ variant was able to rescue this phenotype (Fig. 2b, c and Supplementary Fig. 2d).

Taken together, DYRK1A activity is required for phosphorylation of TOM70 at position Ser91 in vivo. Disturbed DYRK1A signaling leads to impairment of mitochondrial structure, reduced respiration rates, and retardation of cell growth. Thus, TOM70 Ser91 phosphorylation by DYRK1A appears to exert an activating role in mitochondrial protein biogenesis required to maintain mitochondrial function.

**DYRK1A inhibition elicits a protective transcriptional response and remodeling of the import machinery.** We next aimed to analyze if INDY-mediated inhibition of DYRK1A would indeed cause an impairment of TOM70-dependent protein biogenesis. We tested this by importing radiolabelled TIM23 precursor, a model substrate of the TOM70-dependent carrier import pathway[24,25,54], into mitochondria that were isolated from cells grown in the presence or absence of INDY. Surprisingly, import of TIM23, monitored via Blue Native PAGE, was only mildly affected (Fig. 3a). When analyzing the TOM70 phosphorylation status, we noticed that INDY treatment had changed the ratio of phosphorylated towards non-phosphorylated TOM70, but had also led to an overall increase in TOM70 protein levels. This increase in TOM70 abundance resulted in similar amounts of phosphorylated TOM70 species in INDY-treated and in non-treated cells (Supplementary Fig. 2b). This maintained level in TOM70 Ser91 phosphorylation would explain why the carrier import pathway was still relatively functional upon INDY treatment (Fig. 3a). When we further analyzed protein levels via standard SDS-PAGE and immunoblotting, we noticed that next to TOM70 also the other two TOM import receptors, TOM22 and TOM20 were increasing upon DYRK1A inhibition, whereas the central translocation pore TOM40 was not affected (Fig. 3b). Therefore, we speculated that DYRK1A inhibition might trigger a transcriptional response that leads to upregulation of *TOMM70* and possibly also other import receptors to mitigate TOM70 impairment. We tested this by qRT-PCR analysis and observed that INDY treatment led to a significant increase in transcript levels of *TOMM70* and *TOMM20* (Fig. 3c). Furthermore, we found that also the transcripts of *DYRK1A*, its isoform *DYRK1B*, and further kinases of the DYRK family (that had been identified with lower scores in our *KinaseFinder* assay (Fig. 1a and Supplementary Fig. 1h)) significantly increased (Supplementary Fig. 3). To expose the impact of TOM70 pS91 on the carrier import pathway, we treated cells with INDY while in parallel inducing *TOMM70* down-regulation via shRNA (see Methods section) to counteract the transcriptional response. With this experimental setup we could now indeed detect decreased levels of phosphorylated TOM70 upon INDY treatment and as a consequence a strong impairment

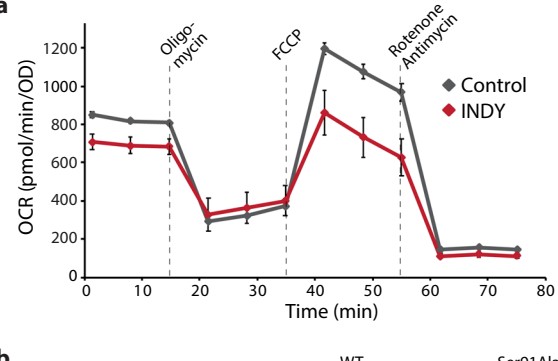

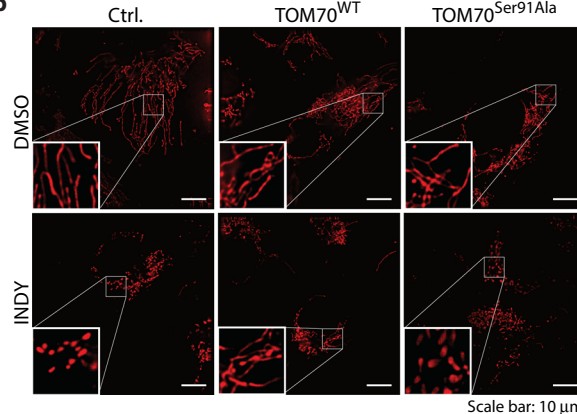

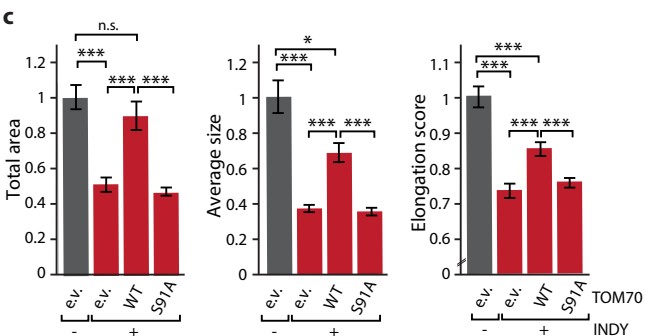

**Fig. 2 Inhibition of DYRK1A activity in vivo affects function and morphology of mitochondria. a** Analysis of oxygen consumption rates (OCR) of cells treated with INDY (10 μM) at basal conditions and after addition of indicated compounds. Control cells were incubated in parallel in the presence of the respective volume of DMSO. Data represent means ± SEM from six independent experiments. **b** U2OS cells cultured on glass bottom plates were treated with 10 μM INDY or DMSO as control for 24 h and stained with MitoTracker Red CMXRos. Where indicated cells were transfected with TOM70$^{WT}$ and TOM70$^{Ser91Ala}$ overexpression plasmids. Each microscopy image includes a magnified inset. Scale bar: 10 μm. **c** Total mitochondrial area, average size and elongation of the samples from **b** were measured. Data were normalized to the respective DMSO control and shown as means ± SEM ($n = 20$ cells). Statistical analysis was performed using a one-way ANOVA followed by a Bonferroni post-hoc test to allow multiple comparisons. Significances are indicated with asterisks: ***$p < 0.001$, **$p < 0.01$, *$p < 0.05$, not significant (n.s.) $p > 0.05$. Numerical source data are reported in the Source Data File.

of the carrier import pathway was revealed (Fig. 3d, e). Notably, mitochondria isolated from DYRK1A knock-out cells revealed reduced levels of TOM70 pS91 and consequently also showed compromised carrier import (Supplementary Fig. 4).

We conclude that inhibition of DYRK1A signaling causes a protective transcriptional response that includes upregulation of

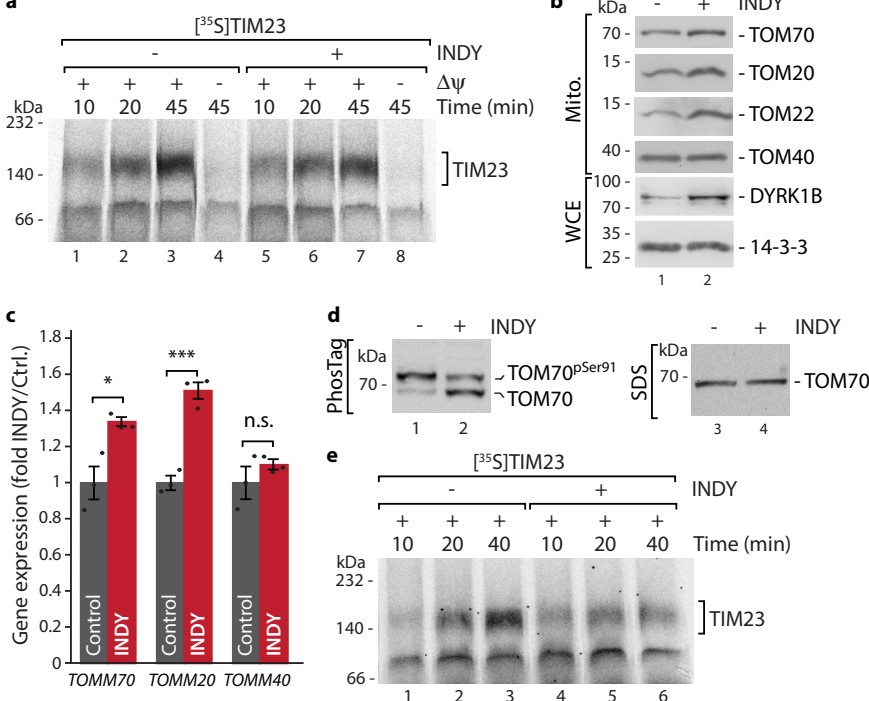

**Fig. 3 DYRK1A inhibition elicits a protective transcriptional response resulting in remodeling of the import machinery. a** Import of radiolabelled TIM23 precursor into isolated mitochondria from cells grown overnight in the presence or absence of INDY (10 μM). Where indicated the membrane potential Δψ was dissipated prior to the import reaction. Samples were lysed in digitonin buffer and subjected to Blue Native PAGE. Imported protein was analyzed by autoradiography. **b** Analysis of protein levels after INDY treatment (samples from **a**) of isolated mitochondria (Mito.) or whole cell exctracts (WCE) via standard SDS-PAGE and immunoblotting. 14-3-3, loading control. **c** Analysis of changes of indicated transcript levels upon DYRK1A inhibition (INDY treatment, 10 μM) by qRT-PCR. $n = 3$, data represent mean ± SEM. Statistical analysis was performed using a two-sided Student's $t$ test to compare between two groups ($p = 0.0132$ (TOMM70); $p = 0.0007$ (TOMM20); $p = 0.1734$ (TOMM40); n.s., not significant). Data are representative of two independent experiments. **d** TOM70 Phos-tag gels (lanes 1 and 2) and SDS-PAGE analysis (lanes 3 and 4) of mitochondria isolated from cells grown overnight in the presence or absence of INDY (10 μM) and, for both conditions, in the presence of TOM70 shRNA. Samples were analyzed by immunodecoration using antisera against TOM70. **e** Import of radiolabelled TIM23 precursor into isolated mitochondria from cells grown overnight in the presence or absence of INDY (10 μM) and in the presence of TOM70 shRNA (as in **d**). Numerical source data and unprocessed immunoblots and autoradiography scans are reported in the Source Data File.

import receptors and further DYRK family members to compensate for loss of DYRK1A activity. This response is likely transient, as comparison of chemical inhibition of DYRK1A with a CRISPR-Cas9 generated DYRK1A knock-out also revealed compromised carrier import. Similarly, attenuation of the transcriptional response via TOMM70 shRNA also results in specific impairment of the carrier import pathway caused by compromised DYRK1A signaling.

**TOM70 Ser91 phosphorylation facilitates interaction with the TOM core complex**. To study how DYRK1A-dependent phosphorylation of TOM70 can influence mitochondrial protein biogenesis at a molecular level we first tested if Ser91 phosphorylation can modulate mitochondrial import of the TOM70 precursor itself. For this we generated radiolabelled precursor variants: TOM70$^{S91A}$, that cannot be phosphorylated and TOM70$^{S91E}$ mimicking the phosphorylated residue. Import into mitochondria was tested by two different assays. (i) Integration of TOM70 into the outer membrane by carbonate extraction (pH 11.5) that separates integral membrane proteins from peripherally associated proteins and (ii) assembly of imported TOM70 by Blue Native PAGE. Analysis of the carbonate-resistant mitochondrial pellets after the import reaction revealed that all $^{35}$S-labeled TOM70 variants integrated into the outer membrane in the same time-dependent manner (Fig. 4a). Analysis on Blue Native PAGE demonstrated that all variants assembled with similar efficiency in

a complex at ~200 kDa probably reflecting formation of TOM70 dimers. Thus, the modification at Ser91 seems not to affect biogenesis of the TOM70 precursor itself (Fig. 4a–c). We next asked if DYRK1A activity might affect the precursor receptor function of TOM70, i.e. modulate its binding efficiency to precursor proteins. For this we employed a direct binding assay for precursor proteins using purified TOM70$_{cd}$ immobilized via deca-His tag to Ni-NTA beads[15,33]. $^{35}$S-labeled precursor proteins are incubated with the TOM70$_{cd}$ and, dependent on their binding affinity to the TOM70 receptor, specifically released from the beads together with TOM70$_{cd}$ upon elution (Supplementary Fig. 5a). To assess the impact of phosphorylated Ser91, immobilized TOM70$_{cd}$ was incubated in the absence or presence of DYRK1A prior to the addition of the radiolabelled precursor proteins. As model substrates of the carrier pathway we tested [$^{35}$S]TIM23 and [$^{35}$S] ANT3 precursors[24,25,54]. Both precursors bound efficiently to the TOM70 receptor domain, however, their interaction was independent of previous phosphorylation of TOM70$_{cd}$ by DYRK1A (Fig. 4d and Supplementary Fig. 5b). [$^{35}$S]Hsp10 as a model substrate of the presequence import pathway[55,56] and [$^{35}$S]GFP as a non-imported control protein were not binding to TOM70$_{cd}$ under any conditions (Fig. 4e, f and Supplementary Fig. 5c). Thus, phosphorylation of TOM70$^{Ser91}$ does not seem to play a role in binding of metabolite carrier precursor proteins. Notably, Ser91 is located close to the transmembrane domain of TOM70 (Supplementary Fig. 5d)[17,30]. Therefore, we speculated that Ser91

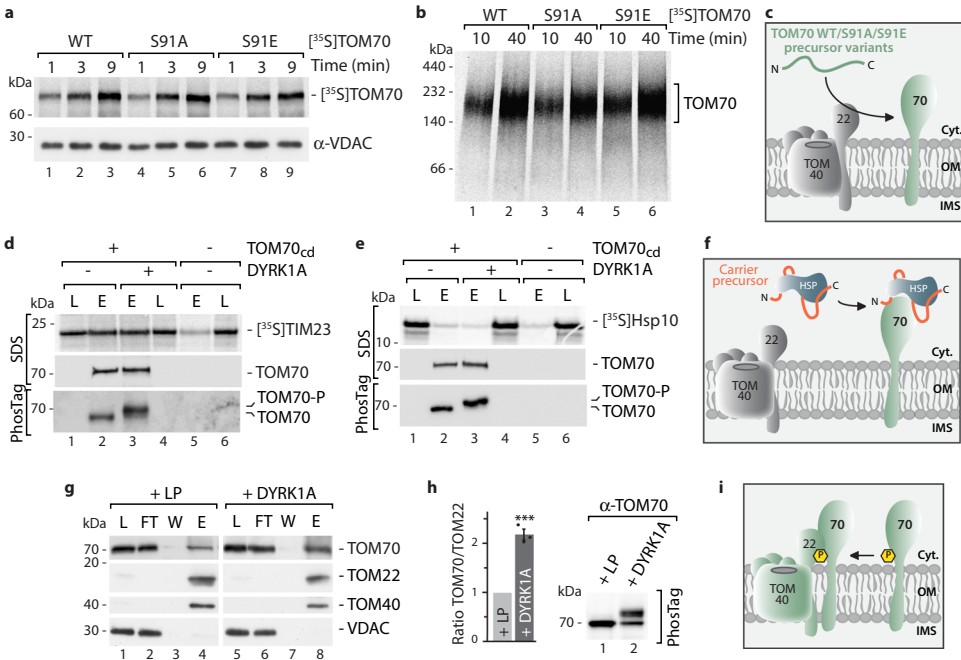

**Fig. 4 Analysis of the role of TOM70$^{Ser91}$ phosphorylation in the TOM complex. a** Indicated [$^{35}$S]TOM70 precursor variants were imported into isolated mitochondria for indicated time followed by protein extraction in 100 mM sodiumcarbonate (pH 11.5). Carbonate-resistant pellet was loaded on SDS-PAGE and imported TOM70 was monitored by autoradiography after transfer onto PVDF membrane. VDAC immunostaining served as loading control. **b** Analysis of import reaction with indicated TOM70 variants via Blue Native PAGE followed by autoradiography. **c** Schematic overview of import assays performed in **a** and **b**. **d** Binding assay for [$^{35}$S]TIM23 precursor (upper panel) to TOM70$_{cd}$ (immobilized via decaHis-tag on Ni-NTA beads) that was incubated before in the absence or presence of DYRK1A. TOM70$_{cd}$ phosphorylation by DYRK1A was controlled via Phos-tag gels (lower panel). L loaded radiolabelled precursor, E elution fraction containing released TOM70$_{cd}$ and radiolabelled precursor in case of specific binding. **e** Binding of [$^{35}$S]Hsp10 precursor to TOM70$_{cd}$ pre-incubated with or without DYRK1A. Assay performed as in **d**. **f** Schematic overview of precursor binding assays performed in **d** and **e**. **g** Immunoprecipitation of TOM complex via TOM22 antibodies from mouse brain mitochondria that were incubated before in the presence of lambda phosphatase (LP; TOM70 dephosphorylated) or DYRK1A (TOM70 phosphorylated). Samples were analyzed by SDS-PAGE followed by immunoblotting with indicated antibodies. L load, FT flowthrough, W wash, E elution. **h** Quantification of TOM70 co-immunoprecipitated with the TOM core complex component TOM22 as shown in **g**. Data represent mean ± SEM from three independent experiments. Statistical analysis was performed using a two-sided Student's *t* test to compare between two groups (*p* = 0.0002). Phos-tag analysis of TOM70 from samples used for immunoprecipitation in **g**. **i** Schematic overview of immunoprecipitation experiment as performed in **g** and **h** to monitor TOM70-TOM core interaction dependent on TOM70$^{pSer91}$ phosphorylation (yellow circle). Numerical source data, unprocessed immunoblots and autoradiography scans are reported in the Source Data File.

phosphorylation could regulate association of the peripheral TOM70 receptor with the core TOM complex, which is required for translocation of the precursors across the outer membrane. To test this hypothesis we incubated isolated mitochondria with lambda phosphatase or DYRK1A to ensure non-phosphorylated or largely phosphorylated TOM70, respectively (Supplementary Fig. 5e). Mitochondria were lysed using a mild non-ionic detergent and the entire TOM complex was enriched by co-immunoprecipitation with TOM22 antibodies. TOM22 is a central component of the TOM core complex, which harbors also the translocation channel TOM40 (29). We found an increase of TOM70 binding to the TOM complex when mitochondria had been treated with DYRK1A prior to the purification (Supplementary Fig. 5f). We further validated the phosphorylation-dependent stimulation of TOM70 interaction with the TOM core complex in mitochondria purified from mouse brain tissue and also detected a significant increase in TOM70 binding to the TOM complex similarly to mitochondria isolated from cultured cells (Fig. 4g, h).

In summary, DYRK1A phosphorylation at Ser91 does not affect TOM70 biogenesis itself or its interaction with incoming precursor proteins. However, phosphorylation of TOM70 Ser91 strongly stimulates the interaction of the peripheral TOM70 receptor with the TOM core complex (Fig. 4i). This interaction is essential for the

transfer of TOM70 receptor-bound precursor proteins to the translocation pore to initiate the import process.

**DYRK1A controls the mitochondrial carrier import pathway via TOM70$^{Ser91}$**. To investigate if TOM70 Ser91 phosphorylation is required to mediate transfer of the precursors to the TOM core complex and thereby initiate translocation across the outer membrane we performed in organello import experiments that allow dissection of the different steps in mitochondrial protein biogenesis. We used TIM23 as model substrate for the TOM70-dependent metabolite carrier import pathway and Hsp10 as model precursor to study the presequence import pathway, which does not depend on TOM70 as import receptor[24,25,54–56] (Fig. 4d, e). We first assessed both import pathways after removal of the TOM70 Ser91 phosphorylation site by lambda phosphatase treatment (LP, Fig. 5a). The carrier import pathway (monitored by membrane-potential (Δψ) dependent assembly into the inner membrane via Blue Native PAGE)[24] was dramatically reduced upon phosphatase treatment confirming the critical role of TOM70 Ser91 phosphorylation in the carrier import pathway (Fig. 5b, c). In contrast, the presequence import pathway was unaffected (Fig. 5d).

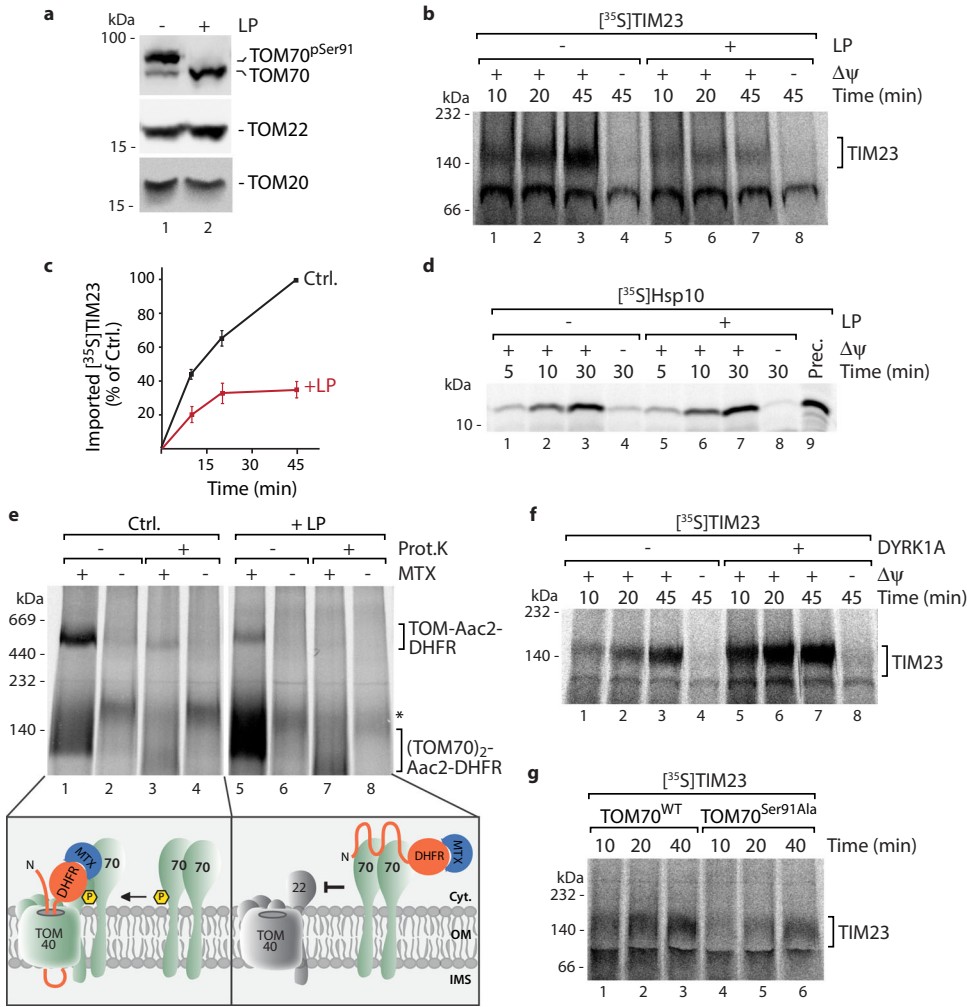

**Fig. 5 TOM70^Ser91 phosphorylation by DYRK1A activates the metabolite carrier import pathway. a** Phos-tag gel analysis of TOM import receptors in mitochondria upon treatment with lambda phosphatase (LP). **b** Import of radiolabelled TIM23 precursor into isolated mitochondria treated with (+) or without (−) lambda phosphatase (LP) as shown in **a**. Where indicated the membrane potential Δψ was dissipated prior to the import reaction. Samples were lysed in digitonin buffer and subjected to Blue Native PAGE. Imported protein was analyzed by autoradiography. **c** Quantification of imported TIM23 from **b**. Ctrl. at 45 min import time was set to 100%. Data represent mean ± SEM from three independent experiments. **d** Import of radiolabelled Hsp10 precursor (Prec.) into isolated mitochondria that were treated with (+) or without (−) lambda phosphatase (LP). Samples were treated with proteinase K after import and analyzed by SDS-PAGE and autoradiography. **e** Import of radiolabelled Aac2-DHFR precursor for 25 min into mitochondria that have been treated as in **a**–**d** in the presence or absence of Methotrexate (MTX). Samples were separated via Blue Native PAGE and analyzed by autoradiography. Proteinase K (Prot. K) treatment reveals accessibility of arrested precursor at TOM complex or TOM70 dimer from the cytosolic side. *, inner membrane assembled Aac2-DHFR is visible in lanes 2 and 4. Cartoons illustrate requirement of phosphorylation for efficient transfer of precursor from TOM70 to the TOM complex. **f** Phosphatase-treated mitochondria were incubated in the presence (+) or absence (−) of DYRK1A followed by import of radiolabelled TIM23 precursor. Reconstitution of TIM23 import was analyzed via Blue Native PAGE and autoradiography. **g** TIM23 import analysis in mitochondria from TOM70 knock-down cells reexpressing TOM70^WT and TOM70^S91A variants. Numerical source data and unprocessed immunoblots and autoradiography scans are reported in the Source Data File.

Import of carrier precursor proteins can be subdivided into different steps: The hydrophobic precursors are bound to cytosolic chaperones, which mediate the initial interaction with the TOM70 receptor. TOM70 binds the precursors and transfers them to the TOM core complex for translocation across the outer membrane. Small Tim chaperones shield the precursor in the intermembrane space and mediate transfer to the TIM22 complex, which inserts the transmembrane helices in the inner membrane in a membrane potential-dependent process[1,3,6,16,17]. To uncover which exact step in the carrier import pathway is affected by removal of TOM70 Ser91 phosphorylation, we imported the ADP/ATP carrier precursor protein fused to a dihydrofolate reductase (DHFR) domain ([35S]Aac2-DHFR)[16]. In the presence of the folate analog methotrexate (MTX) the DHFR

domain is tightly folded so that the entire precursor construct arrests at the TOM complex as translocation across the import channel is blocked. Analysis by Blue Native PAGE revealed that in non-treated mitochondria, in which TOM70 Ser91 is phosphorylated, the Aac2-DHFR precursor accumulates at the TOM complex in the presence of MTX (Fig. 5e, lanes 1–4). This illustrates that TOM70 interacts with the TOM core and transfer of the precursor across the TOM pore is initiated. In contrast, phosphatase treatment of mitochondria prior to the import reaction results in arrest of the Aac2-DHFR predominantly at the TOM70 receptor dimer demonstrating that it cannot be transferred to the TOM core complex due to ablation of the TOM70-TOM-interaction by loss of the TOM70 Ser91 phosphorylation (Fig. 5e, lanes 5–8).

Finally, we wondered if the carrier import pathway can be reactivated by DYRK1A-mediated re-phosphorylation of TOM70[Ser91] (Figs. 1d–f and 4h, and Supplementary Fig. 1f). Indeed, phosphorylation of this single position at TOM70 by DYRK1A led to a vigorous recovery of TIM23 import in phosphatase-treated mitochondria (Fig. 5f). To directly pin the activity of the TIM23 import pathway to the TOM70[Ser91] site we expressed TOM70[WT] or the TOM70[S91A] variant in cells in which expression of endogenous TOM70 had been reduced by shRNA induction (Fig.3d, e). Analysis of TIM23 assembly revealed that the carrier import pathway was strongly impaired in mitochondria isolated from cells expressing the TOM70[S91A] variant compared to TOM70[WT] cells (Fig. 5g and Supplementary Fig. 6).

Taken together, DYRK1A-dependent phosphorylation of TOM70[Ser91] emerges as central modification at the human mitochondrial import machinery critically required for activation of the metabolite carrier import pathway.

## Discussion

Our study identifies a regulatory mechanism of the human mitochondrial import machinery by a cytosolic protein kinase. The unexpected interplay between mitochondrial protein biogenesis and the DYRK1A kinase, which has not been linked to mitochondria before, underscores the necessity of unbiased, global approaches to uncover regulatory principles in mitochondrial and cellular biology. Notably, only a very limited fraction of the human kinome is investigated in detail while for hundreds of kinases the biological function remains enigmatic[36]. Our study shows that a systematic approach like the *KinaseFinder* platform can lead to unforeseen biological links. Impairment of DYRK1A signaling leads to compromised mitochondrial structure and function. DYRK1A exclusively targets the Ser91 position in the peripheral TOM70 receptor and the implications of this phosphorylation on the import process were identified on the molecular level. The modification did not influence biogenesis of the TOM70 receptor itself and unexpectedly did also not impact on its precursor receptor function. In contrast, we found that Ser91 phosphorylation is strongly required to stimulate the transfer of precursors from the receptor-bound state to the translocation pore (Fig. 6). Notably, exactly this step, the transfer of precursors from the TOM70 receptor to the translocation pore, which requires the precursor in a chaperone-bound state, has previously been described as being ATP-dependent. It has been proposed that ATP hydrolysis is required for the release of chaperones from the precursor[17,57,58]. However, our findings offer an exciting alternative explanation, that the ATP dependency is mediated by the protein kinase DYRK1A discovered in this study.

The importance of DYRK1A signaling for functional mitochondria is further underscored by discovery of a protective transcriptional response upon DYRK1A inhibition that includes upregulation of components of the mitochondrial import machinery and of several members of the DYRK family and related kinases, which may act as a safety backup to secure TOM70 Ser91 phosphorylation. It therefore appears that preservation of a functional carrier import pathway via DYRK kinases is an essential process for cell life. Our findings open here an avenue in the converging fields of mitochondrial biogenesis and signal transduction. How the protective transcriptional response discovered in this study is triggered and controlled will be an important research focus in succeeding investigations. Similarly, future studies will be required to link cell-, tissue-, or organ-specific activities of DYRK1A with the necessity to control the mitochondrial import machinery.

Finally, the identification of DYRK1A as a TOM kinase uncovers a connection between DYRK1A signaling and mitochondria. This link opens up the exciting possibility to investigate and decipher the contribution of mitochondrial dysfunction in the many clinical aspects of dysfunctional DYRK1A including *DYRK1A*-related syndrome, Autism Spectrum Disorder, and Down syndrome[37–41].

## Methods

**Tissue culture and mice.** For in vivo and in organello experiments we used human embryonic kidney cells (HEK293T or Flp-In T-REx[TM] 293; Thermo Fisher), human bone osteosarcoma epithelial cells (U2OS, Invitrogen), or mouse brain tissue (C57BL6/N). Cells were cultured in DMEM (Gibco) containing 10% (v/v) fetal bovine serum (FBS; Sigma), 2 mM L-glutamine (Sigma), and 4.5 g/l glucose. Cells were grown at 37 °C in a humidified incubator with 5% $CO_2$. C57Bl6/N mice were bred and maintained in the animal facilities of the University Medical Center Freiburg according to institutional guidelines and in accordance with the German law for animal protection. Mice were kept in specific pathogen-free conditions, fed with standard diet and had access to food and water ad libitum. Temperature in the animal facility was 21–23 °C with 45–60% humidity. A day/night rhythm of 12 h light and 12 h darkness was maintained. Tissue sampling from sacrificed mice was approved by the government commission for animal protection and the local ethics committee (University medical center of Freiburg University; X-18/10 C).

**Generation of cell lines.** For shRNA-mediated depletion of TOM70 the TRIPZ inducible lentiviral non-silencing shRNA system was used (Horizon Discovery, mature antisense: TTATCATACATTTCATCAG, clone ID: V2THS_95541; RHS4743 for control non-silencing). For generation of TRIPZ shRNA lentivirus, $1.6–2 \times 10^6$ HEK293T cells were seeded into 10 cm tissue culture dishes. On the following day, medium was changed and cells were transfected with Polyethylenimine (PEI, Sigma-Aldrich). PEI transfection was performed in a 1:3 DNA-PEI mixture in DMEM without supplements. The mixture was incubated at room temperature for 20 min and added drop-wise to the cells (1/3 of the total medium volume). Lentivirus containing supernatant was harvested after 48 h, and filtered with a 0.2-μm syringe filter. For viral transduction, $5 \times 10^5$ HEK293T cells were

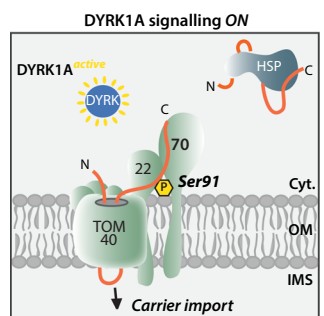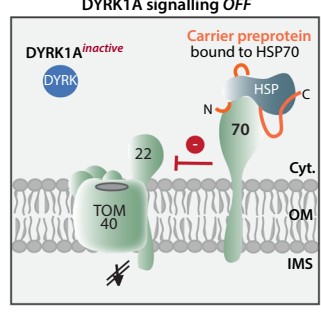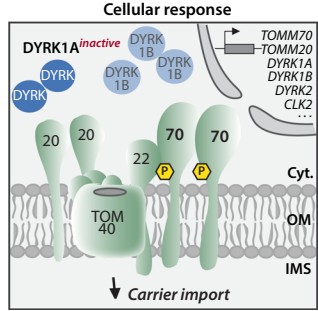

**Fig. 6 Schematic model depicting the role of DYRK1A kinase as critical activator of the carrier import pathway in human mitochondria.** DYRK1A-mediated phosphorylation of TOM70[Ser91] stimulates docking of the precursor bound import receptor to the central translocation pore to enable import of carrier proteins. Upon inhibition, DYRK1A fails to phosphorylate TOM70[Ser91] and protein import is impaired. A protective response secures the carrier import pathway by upregulation of TOM70 and further import receptors as well as backup kinases which can also phosphorylate TOM70[Ser91] to compensate for the lack of DYRK1A.

seeded in six-well plates. Filtered shRNA lentivirus containing supernatant and fresh DMEM containing 10% FBS and 8 µg/ml polybrene (Sigma) were mixed in a 1:1 ratio and added to the cells on the next day. After 24 h, medium was replaced by fresh full DMEM. Afterwards, full DMEM containing 1 µg/ml puromycin (Sigma) was used for selection. One µg/ml doxycycline hydrochloride (Sigma) was used for induction of *TOMM70* gene silencing.

To induce transient overexpression of TOM70 we cloned the cDNA of human *TOMM70* WT and Ser91Ala variant into pMIG plasmids. HEK293T cells were then transfected with pMIG-TOM70 or pMIG-TOM70$^{S91A}$ using PEI as described above and harvested after 24 h. For in vivo fluorescence microscopy U2OS cells were transfected with the same vectors using FuGene (Promega) as described by the manufacturer in 27 mm glass base dishes (Thermo Scientific) and harvested after 48 h.

CRISPR/Cas9 genome editing was used to generate a DYRK1A knockout in the Flp-In T-REx 293 cell line (Thermo Fisher). The required plasmid pSpCas9(BB)-2A-GFP (PX458) was a gift from F. Zhang (Addgene plasmid, #48138)[59]. The guide RNA sequence (GAGAAACACCAATTTCCGAG) targeting exon 6 of human *DYRK1A* was cloned into pSpCas9 (BB)-2A-GFP vector. Flp-In T-REx 293 cells were transfected using Lipofectamine 2000 (Invitrogen). 5 h after transfection, GFP-positive cells were single-cell sorted into 96-well plates by FACS. Clones were expanded and screened for DYRK1A expression via immunoblotting (all primary antibodies used in this study are listed in Supplementary Table 1). For the genomic verification of the DYRK1A knockout cell line, the targeted exon 6 was PCR-amplified from genomic DNA isolated from individual clones with Quick-gDNA MiniPrep (Zymo Research). PCR products were ligated into pCR'-Blunt expression vector (Thermo Fisher) and indel mutations were verified by sequencing.

**Isolation of mitochondria**. Mitochondria were isolated as previously described by Johnston et al.[60]. Eight-weeks-old mice were euthanized by cervical dislocation and brain tissue removed. Subsequently, tissues were weighed and cut into small pieces. For isolation of mitochondria from HEK293T cells, cells were harvested. Tissue and cell pellets were resuspended in solution A (220 mM mannitol, 70 mM sucrose, 20 mM HEPES-KOH, pH 7.6, 1 mM EDTA, 0.5 mM PMSF, and 2 mg/ml BSA) and subsequently homogenized using a glass potter. Cellular debris was removed by centrifugation at 800 × *g* for 5 min at 4 °C and mitochondria were pelleted from the supernatant by further centrifugation at 10,000 × *g* for 15 min at 4 °C. Crude mitochondrial pellet was resuspended in solution B (solution A without BSA) and again centrifuged at 800 g for 5 min (4 °C). Mitochondria were then pelleted by a 10,000 × *g* spin for 15 min and resuspended in import buffer (described below). Protein concentrations were determined by Bradford assay and samples stored at −80 °C or directly used for functional assays. For in vitro experiments we used soluble cytosolic domains of the human import receptors TOM20 (residues 35–145), TOM22 (residues 2–80), and TOM70 (residues 70–608), which were either generated by expression and affinity purification in *E. coli* (TOM20 and TOM70, each with N-terminal deca-Histidine tag) or directly synthesized in case of TOM22 (GeneCust).

**Expression and purification of soluble TOM receptor domains and Frataxin**. cDNAs encoding cytosolic domains (cd) of human TOM70, TOM20, and Frataxin (FXN) (full length) were cloned into pET10N vector[61] providing a deca Histidine tag at the N-terminus of the protein. Point mutations for generating the TOM70$_{cd}$$^{T85A}$, TOM70$_{cd}$$^{S91A}$, and TOM70$_{cd}$$^{S110A}$ variants were introduced by site-directed mutagenesis. Vector constructs were transformed into *E. coli* and cells were grown at 37 °C until an OD$_{600}$ of 0.6. Expression was induced by addition of 1 mM IPTG. After growth for additional 3 h, 40 ml of the culture was harvested and pelleted cells were resuspended in extraction buffer (20 mM Tris/HCl, pH 8.0, 4 mM MgCl$_2$, 0.5 mM PMSF) and incubated with 1 mg/ml lysozyme and DNase I for 30 min on ice. Lysate was cleared at 10,000 × *g* for 10 min at 4 °C. The supernatant was incubated with Ni-NTA resin beads (Qiagen) and washed three times with washing buffer (20 mM Tris/HCl, pH 8.0, 50 mM Imidazole, 200 mM NaCl, 1 mM β-mercaptoethanol). Proteins were eluted with elution buffer (20 mM Tris/HCl, pH 8.0, 350 mM Imidazole, 100 mM NaCl, 1 mM β-mercaptoethanol). Grades of protein purity were analyzed by SDS-PAGE and Coomassie brilliant blue staining.

**Phosphorylation assays**. For in vitro phosphorylation, purified receptor domains (2 µg) were incubated in 30 µl ST buffer (10 mM Tris/HCl, pH 7.2, 250 mM sucrose, 2 mM MgCl$_2$, and 1 mM PMSF) containing 5–10 mM ATP and 1 µl of the respective protein kinase (purchased from Thermo Scientific, NEB or ProQinase). Reactions were incubated for 45 min at 30 °C (gentle agitation at 500 rpm) and subsequently stopped by addition of Laemmli buffer. Samples were analyzed by Phos-tag gels and immunoblotting.

For in organello phosphorylation, isolated mitochondria (50 µg) were incubated in 50 µl kinase assay buffer (10 mM Tris/HCl pH 7.4, 250 mM sucrose, 10 mM MgCl$_2$, 2 mM PMSF, 1x PhosStop (Roche), 1x *EDTA* free protease inhibitor cocktail (Roche), and 1x Protein Metallo Phosphatase (PMP) buffer (NEB)). For initial dephosphorylation 1 µl lambda phosphatase (LP, NEB) was added and incubated for 45 min at 37 °C (gentle agitation at 550 rpm). Mitochondria were washed twice with sucrose buffer (10 mM HEPES/HCl, pH 7.6, 0.5 M sucrose) and

resuspended in assay buffer containing 5–10 mM ATP and 1 µl of the respective protein kinase (NEB, ProQinase or Thermo Fisher). After incubation for 45 min at 37 °C (gentle agitation at 550 rpm) mitochondria were washed with sucrose buffer and resuspended in Laemmli buffer. Samples were analyzed by Phos-tag gels and immunoblotting.

**Kinase inhibition assays**. HEK293T or U2OS cells (for in vivo fluorescence microscopy) were cultured in full DMEM media as described above in the presence of 10 µM INDY (Tocris Bioscience; 10 mM stock in DMSO) for different periods of time (overnight, 24 h, 48 h, or 72 h). Cells were subjected to functional assays as described below. For cell growth 5 × 10$^4$ cells were seeded and incubated in the presence of 10 µM INDY or DMSO as control. Cell numbers were determined using a Neubauer cell counting chamber after 48 h.

To assess inhibition by INDY in vitro, DYRK1A kinase was incubated with different concentrations of INDY (2–10 µM) for 30 min in 30 µl ST buffer containing 5–10 mM ATP. Purified TOM70$_{cd}$ was added and samples further incubated for 45 min at 30 °C (gentle agitation at 500 rpm). Reactions were stopped by addition of Laemmli buffer. Samples were analyzed by Phos-tag gels and immunoblotting.

**In vitro transcription/translation and in organello import of precursor proteins**. Radiolabelled precursor proteins (human TOM70$^{WT}$, TOM70$^{S91A}$, TOM70$^{S91E}$, TIM23, ANT3, Hsp10 (yeast), Aac2-DHFR, and GFP) were synthesized by in vitro transcription/translation in the presence of $^{35}$S-methionine using the rabbit reticulocyte lysate system (Promega)[34,62]. For import assays, isolated mitochondria and radiolabelled precursor proteins were mixed in import buffer (250 mM sucrose, 5 mM magnesium acetate, 80 mM potassium acetate, 10 mM sodium succinate, 20 mM HEPES-KOH, pH 7.4) supplemented with 1 mM DTT and 5–10 mM ATP and incubated for various time points at 37 °C. For disruption of the membrane potential Δψ, antimycin A (8 µM), valinomycin (1 µM), and oligomycin (20 µM) were added prior to the import reaction. To remove endogenous phosphorylation sites, isolated mitochondria were incubated with LP for 30 min in ST buffer before the import reaction. For folding of the DHFR domain to block Aac2-DHFR precursor translocation methotrexate (20 µM) was added prior to the import reaction[16]. For removal of non-imported precursor proteins samples were treated with 20 µg/ml Proteinase K. Mitochondria were re-isolated by centrifugation at 10,000 × *g* for 10 min at 4 °C and samples were analyzed by SDS-PAGE and autoradiography[62]. For analyzing protein assembly, mitochondrial pellets were lysed in digitonin buffer (0.4–1% (w/v) digitonin, 0.5 mM EDTA, 10% (w/v) glycerol, 50 mM NaCl, 20 mM Tris-HCl, pH 7.4) and incubated on ice for 15 min. After clarifying spin at 20,000 × *g* for 5 min at 4 °C, supernatant was loaded on gradient Blue-native PAGE (4–13% or 6–16% (w/v) acrylamide) followed by autoradiography or immunoblotting[16,34,60]. Images were processed using ImageJ (1.52) software.

**Phos-tag electrophoresis**. Zn$^{2+}$-Phos-tag Bis-Tris gel electrophoresis was performed as described previously[48]. Briefly, discontinuous 8% or 12.5% SDS-PAGE was prepared with the modification that 12.5 µM Phos-tag reagent and 25 µM ZnCl$_2$ were added to the separation gel mix before polymerization (50 µM Phos-tag and 100 µM ZnCl$_2$ for analysis of TOM20 and TOM22 phosphorylation). Electrophoresis was performed for 3–4 h at 35 mA and 600 V. Afterwards, wet transfer process (Biorad system) was used for protein transfer onto a PVDF membrane. Proteins were detected via immunodecoration.

**Precursor binding assay**. Purified receptor domains with N-terminal deca-His tags or control proteins that had been incubated in the presence or absence of DYRK1A as described above were resuspended with Ni$^{2+}$-NTA resin equilibrated with assay buffer (20 mM Imidazole, 10 mM KCl, 10 mM MOPS/KOH, pH 7.2, 1% (w/v) BSA). After incubation for 30 min (4 °C) resin was transferred to 1 ml Mobicol mini column. [$^{35}$S]-labeled precursor or control proteins were diluted in assay buffer (up to max. 5% (v/v) rabbit reticulocyte lysate was used) and added to the resuspended resin. After incubation for 30 min at 30 °C resin was pelleted and washed three times with assay buffer without BSA. Bound proteins were eluted with Laemmli buffer and analyzed by SDS-PAGE, autoradiography, and immunoblotting.

**Co-immunoprecipitation of the TOM complex**. Mitochondria were isolated from mouse brain tissue or HEK293T cells and incubated either with DYRK1A or Lambda phosphatase as described above to achieve a maximally phosphorylated or non-phosphorylated state of TOM70. Samples were then lysed in 0.5% (w/v) Digitonin buffer with 1x PhosStop (Roche) and 1x EDTA free protease inhibitor cocktail (Roche). Lysates were cleared by centrifugation at 20,800 × *g* for 10 min at 4 °C. Supernatants were incubated with pre-coupled anti-TOM22 serum to Protein A-Sepharose beads (GE Healthcare). After incubation for 40 min at 4 °C, the resin was collected by centrifugation and washed six times with 0.1% digitonin buffer. Bound proteins were eluted with Laemmli buffer and subjected to SDS-PAGE or Phos-tag gels and immunoblot analysis.

**Measurement of respiratory activity**. Mitochondrial respiratory activity was measured using a XF96 Extracellular Flux analyzer (Seahorse Bioscience). Cells were seeded at a density of $2 \times 10^4$ cells/well on a poly-D-lysine coated plate one day before the measurement. Basal levels of oxygen consumption rate (OCR) and OCR in the presence of electron transport chain inhibitors and uncouplers (2 µM oligomycin, 0.3 µM carbonyl cyanide-4-(trifluoromethoxy) phenylhydrazone, 1 µM antimycin A and 1 µM rotenone) were measured. OCR values were normalized to the protein content (determined after the measurement by Bradford assay).

**In vivo fluorescence microscopy**. U2OS cells were grown in 27 mm glass base dishes (see above) and incubated in the absence or presence of INDY (10 µM) for 24 h. Prior to visualization, cells were stained for 30 min with 50 nM MitoTracker Red CMXRos (Thermo Scientific) in culture media. Fluorescent microscopy images were recorded with a DeltaVision Ultra High Resolution Microscope with UPlanSApo 100x/1.4 oil Olympus objective, using a sCMOS pro.edge camera at 37 °C. Raw fluorescent microscopy images were deconvolved at the DeltaVision microscope using SoftWorx deconvolution plugin. Quantification of morphological parameters of mitochondria was performed with the ImageJ 1.52 software (Wayne Rasband, National Institutes of Health, USA), using the Analyze Particles functionality[63]. Elongation was calculated as the inverse of circularity. At least 50 cells were analyzed for each condition.

**In vitro γ-[$^{18}O_4$]-ATP kinase assay and MS analysis**. Isolated mitochondria were dephosphorylated by LP treatment as described above. For re-phosphorylation 40 ng DYRK1A/µg mitochondria was added and incubated in kinase assay buffer in the presence of 6 mM γ-[$^{18}O_4$]-ATP (Cambridge Isotope Laboratory), 0.5 mM sodium orthovanadate, and 2.5 mM sodium fluoride for 60 min at 37 °C with gentle agitation at 550 rpm. Samples were washed with sucrose buffer and centrifuged at $10,000 \times g$ for 10 min at 4 °C. Pellets were snap-frozen and stored at −80 °C. Samples were lysed in 8 M urea and transferred to 10 kDa MW-cutoff filters (Sartorius). Trypsin digestion was performed overnight according to Filter Aided Sample Preparation (FASP)[64]. On the next day, the peptides were eluted twice with 100 µL of 50 mM ammonium bicarbonate. The eluates were acidified with 6 µl of 50% TFA and lyophilized. Dry peptides were resuspended in 200 µl 80% acetonitrile/1% TFA and incubated with 2 mg of TiO₂ (GL Sciences) for 30 min, which were equilibrated with 300 mg/ml lactic acid in 80% acetonitrile/1% TFA[65]. TiO₂ beads were washed with 200 µl of 10% acetonitrile/1% TFA, 200 µl of 80% acetonitrile/1% TFA, and 50 µl LC-MS grade water. Phosphopeptides were eluted with 50 µl of 1% ammonia in 20% acetonitrile and 50 µl of 1% ammonia in 40% acetonitrile. Flow-throughs were measured for non-phosphopeptide analyses.

LC-MS/MS measurements were performed on a QE HF-X mass spectrometer coupled to an EasyLC 1200 nanoflow-HPLC (all Thermo Scientific) essentially as described before[64]. The mass spectrometer was operated in the data-dependent mode; after each MS scan (mass range $m/z = 370–1750$; resolution: 120,000) a maximum of twelve MS/MS scans were performed using a normalized collision energy of 25%, a target value of 5,000 and a resolution of 30,000. MS raw files were analyzed using MaxQuant (version 1.6.2.10)[66] using a Uniprot full-length Homo sapiens (Human) database (March 2016) and common contaminants such as keratins and enzymes used for in-gel digestion as reference. Carbamidomethylcysteine was set as fixed modification and protein amino-terminal acetylation, serine-, threonine-, and tyrosine- heavy phosphorylation, and oxidation of methionine were set as variable modifications. The MS/MS tolerance was set to 20 ppm and three missed cleavages were allowed using trypsin/P as enzyme specificity. Peptide, site, and protein false discovery rate based on a forward-reverse database were set to 0.01, minimum peptide length was set to 7, the minimum score for modified peptides was 40, and minimum number of peptides for identification of proteins was set to one, which must be unique.

**Radiometric filter binding assay**. In vitro kinase screening was performed for 339 human protein kinases (245 Ser/Thr kinases and 94 Tyr kinases) using 96-well plates and 5 µg purified human TOM70$_{cd}$ or TOM70$_{cd}$S91A protein per well in the presence of 1 µM γ-[$^{33}P$]-ATP ($9.08 \times 10^{05}$ cpm per well) according to the manufacturer's protocol[46]. Screening was performed by ProQinase (Freiburg, Germany). Values for each kinase were given as counting of cpms of incorporated $^{33}P_i$ and autophosphorylation activity (in the absence of TOM70$_{cd}$ or TOM70$_{cd}$S91A) was subtracted. A list of the tested kinases is available at https://www.proqinase.com/sites/default/files/public/uploads/KinaseFinder/pq_list_kinasefinder_339_sty_kinases_v03.pdf. The two top score kinases for TOM70$_{cd}$, DYRK1A and DYRK1B, both with activity ratios >70 were further validated by titrating the substrate from 0, 1.25, 2.5 and 5 µg to reveal a substrate concentration-dependent kinase activity.

**Identification of TOM70 phosphorylation sites in HEK293T cells**. Isolated mitochondria were lysed in three consecutive cycles of snap-freezing (N₂ liquid), thawing and 1 min incubation in an ultrasonic bath in presence of 4% SDS in 50 mM Tris/HCl and 150 mM NaCl, pH 7.8 including 1x PhosStop (Roche) and 1x EDTA free protease inhibitor cocktail (Roche). Proteins were reduced by adding DTT to a final concentration of 10 mM and incubation for 30 min at 56 °C

followed by alkylation via iodoacetamide using a final concentration of 20 mM for 30 min in the dark.

Proteins were precipitated by adding 9 volumes of ice-cold ethanol and incubation at −40 °C for 1 h. Precipitated proteins were spun down at $12,000 \times g$ for 40 min at 4 °C and pellets were washed using 100 µl of ice-cold acetone. Protein pellets were solubilized in 6 M guanidinium-HCl and the protein concentration was determined using a bicinchoninic acid assay (Pierce, Thermo Scientific). Prior proteolytic digestion, samples were diluted 20-fold using 50 mM ammonium bicarbonate, 1.5 mM CaCl₂. Trypsin (Sequencing grade, Promega,) was added in a ratio of 1:20 (trypsin:protein, w/w) and samples were incubated for 13 h at 37 °C under slight agitation. The digestion was stopped by addition of FA to a final concentration of 1% and peptides were purified using SPEC-C18 AR cartridges (Agilent). Digestion quality was controlled via a monolithic column-HPLC, as described previously[67].

Electrostatic repulsion-hydrophilic interaction liquid chromatography (ERLIC) was performed similarly to Loroch et al.[68] using an Ultimate 3000 HPLC system (Thermo Scientific) equipped with a PolyWAX (weak anion exchange) column (4.6 mm ×;100 mm, 5 µm, 300 Å, Poly LC, Columbia) eluted with a binary gradient of 20 mM sodium methylphosphonic acid, 70% acetonitrile, pH 2 (ERLIC buffer A) and 200 mM triethylammonium phosphate, 60% ACN, pH 2 (ERLIC buffer B). Separation was conducted at a flow rate of 1 mL/min starting with 100% A for 10 min followed by a linear increase to 100% B over 10 min followed by 100% B for 5 min. 24 fractions of 1 min were collected. To remove acetonitrile and residual ERLIC buffer salts, fractions were dried, reconstituted in 0.1% TFA, subjected to RP-SPE using Hypersep SpinTips (Thermo Scientific) followed by LC-MS.

LC-MS was done using an LTQ Orbitrap Velos Pro mass spectrometer online coupled to a U3000 RSLC nanoHPLC equipped with an Acclaim PepMap trap-column (100 µm ×;2 cm, 5 µm particles, 100 Å pores) and an Acclaim PepMap main column (75 µm ×;15 cm, 2 µm particles, 100 Å pores). After 5 min the trap column was switched in-line and peptides were separated on the main column at a flow rate of 250 nl/min using a linear acetonitrile gradient (2.5–30%) in presence of 0.1% FA. Gradient lengths were adjusted to the complexity of the fractions: 160 min for fraction 1–3; 100 min for fraction 4; 50 min for fractions 5–15; and 20 min for fraction 16–24. The column effluent was introduced to the MS using an NSI source equipped with a 10-µm PicoTip emitter (New objective) using a voltage of 1.5 kV. The MS was operated in data-dependant acquisition mode with a survey scan acquired in the orbitrap at a resolution R = 60,000 followed by up to 15 MS/MS of the most intense precursor ions with a charge of +2 to +4 in the linear ion trap using collision-induced dissociation (top15 CID). The maximum ion injection time was 100 ms and the AGC target values were set to $10^6$ and $10^4$ for MS and MS/MS, respectively. Ions were excluded from re-fragmentation for 20 s (dynamic exclusion).

For data analysis, raw-files were analyzed using Proteome Discoverer v1.4. Database search was conducted using Mascot v2.4 against Uniprot, taxonomy human (www.uniprot.org, 11-2016, 20,120 target sequences). DB search parameters were: trypsin with two missed cleavage sites; carbamidomethylation of cysteine as fixed modification; oxidation of methionine, N-terminal acetylation and phosphorylation of serine, threonine, and tyrosine as variable modification. Error tolerances were set to 10 ppm and 0.5 Da for precursor and fragment ions, respectively. phosphoRS v3.1 was used to assign phosphosite localization probabilities[69] and results were filtered to meet a 1% false discovery rate (on PSM level) using the Percolator 2.04 node[70]. Phosphosites were considered identified if the phosphoRS site probability was ≥90%. We identified 212 phosphorylation sites derived from 101 mitochondrial proteins (filtered against MitoCarta 2.0) of which three phosphopeptides derived from TOM70 (Supplementary Fig. S1C). The mass spectrometry proteomics data have been deposited to the ProteomeXchange Consortium via the PRIDE[71] partner repository with the dataset identifier PXD019520.

**RNA isolation and qRT-PCR**. Total RNA was isolated using the RNeasy Mini Kit (Qiagen) with DNase treatment to omit DNA contaminations. cDNA was synthesized with High-Capacity cDNA kit (Applied Biosystems) from 2 µg RNA. PCR amplifications and detections were performed with CFX Connect Real-time PCR detection system (Bio-Rad) using SsoAdvanced Universal SYBR Green Supermix (Bio-Rad). β-Actin (ACTB) was used as housekeeping gene for normalization. Relative mRNA levels were calculated with the delta-delta Ct method. A complete list of primers is provided in Supplementary Table 2.

**Statistics and reproducibility**. Data shown represent means ± standard error of the mean (SEM). Statistical details of each experiment can be found in the figure legends. A two-sided Student's $t$ test was applied to compare between two groups. A one-way ANOVA followed by a Bonferroni post-hoc test was used to allow multiple comparisons. The level of significance is indicated in the figures with asterisks (***$p < 0.001$, **$p < 0.01$, *$p < 0.05$, not significant (n.s.) $p > 0.05$) and is provided in the figure legends as exact p-value as obtained by the indicated statistic test. Multiple independent experiments were carried out for each experiment and all biochemical experiments were replicated at least three times and obtained the same results. Respiratory activity was determined in two biological replicates (six technical replicates each) and obtained the same results. Gene expression analysis (three replicates in each measurement) and in vivo microscopy analyses were

performed from two biological replicates in independent experiments and were reproducible.

**Reporting summary**. Further information on research design is available in the Nature Research Reporting Summary linked to this article.

## Data availability

MS/MS data are deposited to the ProteomeXchange Consortium via the PRIDE partner repository (https://www.ebi.ac.uk/pride) with the dataset identifier PXD019520. Source data and uncropped versions of gels, autoradiography scans and immunoblots are provided in the Source Data File. Further data and resources from this study are available from the corresponding author upon reasonable request.

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

## Acknowledgements

The authors thank Dr. Sabrina Büttner and Dr. Frank Totzke for scientific discussion and Lisa Myketin and Cornelia Schumbrutzki for expert technical assistance. Work included in this study has also been performed in partial fulfillment of the requirements for the doctoral theses of C.W., H.P. and C.K. at the University of Freiburg. This work was supported by the Deutsche Forschungsgemeinschaft (DFG), under Germany's Excellence Strategy (CIBSS - EXC-2189 - Project ID 390939984 to C.M., Cl.K. and F.N.V.), the Heisenberg-Professorship BR 3662/5 (to T.B.), the RTG 2202 and ME 1921/5-1 (to C.M.), the SFB1381 (Project-ID 403222702; to C.M., Cl.K. and F.N.V.), the SFB1177 (Project-ID 259130777; to Cl.K.), project ID 409673687 (to Cl.K.) and the Emmy-Noether Programm of the Deutsche Forschungsgemeinschaft (to F.N.V.). Further funding was received from DST-SERB, India (to A.M.), the German Cancer Consortium (DKTK, L627), the Motivate MD college (to H.P.), the Swiss National Science Foundation and the canton of Fribourg (to J.D.), from the European Research Council (ERC) under the European Union's Horizon 2020 research and innovation program under grant agreement No 769065 and from the European Union's Horizon 2020 research and innovation program under grant agreement No 765912 (to Cl.K.). This work reflects only the authors' view and the European Union's Horizon 2020 research and innovation program is not responsible for any use that may be made of the information it contains. Furthermore, this research project was funded by Ministry of Science and Culture of Lower Saxony and Volkswagen Foundation No 762-12-9/19 (ZN3457) (to S.D.).

## Author contributions

C.W., A.M., T.S., V.M., R.E., C.K., P.S.M., Z.H., A.A., S.L., F.A.S., A.SCH., D.P.H., H.P., S.M., S.D. and T.B. performed the experiments. C.W., A.M., T.S., V.M., R.E., A.S., C.K., J.B., J.D., S.D., C.L.K., T.B., F.N.V. and C.M. designed experiments, analyzed and interpreted the data. C.W., A.M., F.N.V. and C.M. developed the project and wrote the manuscript. C.M. coordinated and directed the project. All authors approved the final version of the manuscript.

## Funding

## Competing interests

The authors declare no competing interests.
