## [Peer Review File · Nature Communications]

REVIEWER COMMENTS

Reviewer #1 (Remarks to the Author):

Mitochondrial surface receptors serve as the interface between the cytosolic and the mitochondrial proteomes. They recognize cytosolic precursor proteins and, in the case of TOM70, recruit chaperones to the mitochondrial surface. TOM70 is a complicated protein with functionally distinct domains (Clamp, Core and C-terminal domain) which binds to a number of different partners, including precursors, chaperones, J-proteins, the endoplasmic reticulum, components of the ubiquitin-proteasome-system as well as Msp1/ATAD1. How these different partners are bound and how the different activities of TOM70 are coordinated is largely unknown.

In this manuscript, Walter and coauthors describe a post-translational modification of human TOM70 that is of crucial relevance for its functionality. Phosphorylation of a serine residue at position 91 strongly increases its activity in the import reaction of the model substrate TIM23. It might also increase the fraction of TOM70 that forms a complex with the TOM translocase. Phosphorylation of serine 91 of TOM70 is mediated by DYRK1A/B proteases. At steady state, TOM70 appears to be predominantly in the active, i.e. phosphorylated state.

This is an exciting study which sheds light on the regulation of an important factor of the mitochondrial import machinery. Given its different functions in the context of mitochondrial protein biogenesis, TOM70 is a highly interesting candidate as regulatory control element. The identification of DYRK1A/B and the relevance of the phosphorylation of serine 91 for mitochondrial biogenesis are very convincingly shown in this study. The molecular details of this regulatory step will have to be further analyzed in the future. However, already this initial discovery study here is of very high quality and will be a significant advance for the field. Some minor points still should be considered.

Specific points

1. The authors introduce INDY as specific inhibitor of DYRK1A/B activity. If their conclusion is correct than INDY-treated cells (or mitochondria) should have reduced levels of TOM70-dependent substrates. It should be straight forward to test this by Western blotting or MS.
2. Does the overexpression of TOM70 suppress the INDY-induced mitochondrial morphology phenotype? This would be expected if the DYRK1A/B role is indeed exclusively explained by its activation of TOM70.
3. The authors nicely show that phosphorylation is not relevant for the ability of TOM70 to bind substrates (in vitro). However, they did not test any relevance for the chaperone-binding Clamp domain which is in direct proximity to the serine 91 residue. However, a (presumably the) critical function of Tom70 is the binding of chaperones via its EEVD-binding Clamp domain. To analyze the effect of serine 91 phosphorylation on chaperone binding is beyond the scope of this initial study, but it should be discussed.
4. The data on the TOM70 phosphorylation by DYRK1A/B are very sound and convincing. However, the evidence of a phosphorylation-induced TOM70 recruitment to TOM22 is not. The latter is just based on a somewhat stronger band in the pulldown shown in Fig. 3g. The increase in the pull-down signals in Fig. 3h is strong from the quantification, but actually, the signals of the band in the +LP sample looks stronger to me than those after the DYRK1A treatment. Since the authors have the INDY inhibitor, it should be straight forward to improve the evidence for the DYRK1A/B-dependent TOM70 recruitment.
5. In the past, crosslinking was very powerful to show the binding of carriers to TOM70 (see for example Fig. 1A of Wiedemann et al, EMBO J 20, 951f). This simple assay could be a powerful basis to provide convincing evidence for the claim that phosphorylation of TOM70 is not required for carrier binding, however crucial for the subsequent TOM70 -> TOM40 transfer. This should be considered.

Minor points:

6. The microscopy image in Fig. 2b is too dark. The mitochondria are hardly visible, at least on my screen
7. Page 5, five lines from bottom: For consistency, Tom70S110A should be TOM70S110A
8. Page 6, top line: gelelectrophoresis should be gel electrophoresis

Reviewer #2 (Remarks to the Author):

Walter et al. performed in vitro profiling of the human kinome and identified that DYRK1A is able to phosphorylate TOM70 with high efficiency. Through combination of LC-MS for phosphorylation site identification and DYRK1A treatment of TOM70 bearing different mutation site, the authors showed that DYRK1A-dependent phosphorylation of TOM70 happens specifically at Ser91. Further experiments showed that Ser91 phosphorylation of TOM70 would increase its binding to the TOM complex. Finally, a simple model about DYRK1A-dependent carrier import pathway was proposed according to these observations.

The authors made an interesting discovery. However, mechanistic details upstream and downstream of this phosphorylation event is necessary to make the story complete. Also, more evidence is needed to support the assumptions made by the authors. I suggest major revision should be made before it meets the publication criteria of Nature Communications.

1. It's not quite accurate to say that "Human TOM70 is nearly quantitatively phosphorylated at Ser91.". As indicated by LC-MS results and the existence of bands containing multiple phosphor groups during phos-tag gel-electrophoresis of untreated mitochondria (Figure 1d), it seems that, while nearly all TOM70 is phosphorylated, Thr85 and Ser110 also contribute to the phosphorylation of TOM70. The band in the Ser91-specific antibody staining cannot rule out the possibility that mono-phosphorylated TOM70 at Thr85 or Ser110 exists. Further evidence is necessary to support the statement that TOM70 is nearly quantitatively phosphorylated at Ser91. Authors may try to use MS-based quantitative method like MRM-MS to explain this.

2. Overall the authors haven't adequately addressed the issue of other kinases that might phosphorylate TOM70 and other substrates DYRK1A can phosphorylate that can interfere with functional assays of TOM70 phosphorylation. For example, while DYRK1A/B received the highest score in the screening assay, HIPK1 is a close second. HIPK1 also belongs to the DYRK family. Have the authors look into HIPK1 as well for potential cross reactivity? As indicated by the authors, DYRK1A is implicated in several diseases and there are multiple reported DYRK1A substrates including MAP1B. The functional assays through inhibition of DYRK1A (e.g., Fig. 2d and Extended Data Fig.2a) do not necessarily attribute to the change of TOM70 phosphorylation.

3. In Figure 2a, Ser91 phosphorylated TOM70 still exists after INDY treatment. Is it due to insufficient inhibition of DYRK1A or that there is other kinase(s) responsible for Ser91 phosphorylation of TOM70? Did the authors try to raise the concentration of INDY or perform a DYRK1A-knockout experiment?

4. Direct evidence, like in vivo TOM70 S91A mutant experiments, connecting loss of Ser 91 phosphorylation of TOM70, and impairment of mitochondria structure and functions is necessary. DYRK1A have many substrates and inhibition of it would lead to down-regulation of phosphorylation of many proteins.

5. Assessment of TOM70 binding with TOM complex was performed using mouse brain mitochondria is not reasonable. Using mitochondria from HEK293T cells is necessary. Data from different species shouldn't be simply combined.

6. It seems that the amount of imported TIM23 increased significantly between 20 and 45 mins without DYRK1A treatment, as shown in Figure 4e, which is contradict to what shown in Figure 4b and 4c. What's the possible reasons?

7. While lambda phosphatase treatment is effective in removing phosphorylation, it is neither substrate-specific nor site-specific. Direct evidence suggesting Ser91 phosphorylation is responsible for TOM70 binding of TOM complex and TIM23 import should be provided, since the dephosphorylation and subsequent phosphorylation of other protein(s) within the mitochondria could be (one of) the cause(s). Besides, was the level of TIM23 import after DYRK1A treatment comparable with that before lambda phosphatase treatment?

8. Whether DYRK1A is the only kinase of TOM70 is not clearly discussed. This criticism is due to the lack of appropriate controls in some of the new key experiments presented. It is necessary to

knockout DYRK1A and show its affection on the phosphorylation of TOM70.

9. In the main text (P5), the author said the phosphopeptide is enriched by ERLIC but in the method part, the phosphopeptide is enriched by TiO₂.

Reviewer #3 (Remarks to the Author):

This is a well done study characterizing the phosphorylation of Ser91 of TOM70 by DYRK1A and showing this regulates TOM70 interaction with the core TOM70 import complex for mitochondrial import of receptor bound precursors. The functional role for DYRK1A phosphorylation and regulation of TOM70 is potentially an important discovery for dysfunctional DYRK1A including DYRK1A-related syndrome, Autism Spectrum Disorder and Down syndrome.

Specific recommendations to improve the manuscript:

1. The kinase screen in Fig.1A identifies DYRK1A/B as having the highest activity ratios in the screen. But several other kinases have very significant activity including HIPK1, PKC-beta, other PKCs and CLK. But in Fig. 1B several kinases are compared to DYRK1A/B for comparison TOM70cd Ser 91 that have very low activity ratios in the screen (CLK3, LIMK, RPS6KA1, CK1-delta). The phosphorylation assay in Fig.1B should be shown for HIPK1, the different PKCs and CLK. It is possible that additional kinases have the ability to phosphorylate TOM70 Ser 91 and regulate functional TOM70 activity. This must be tested.

2. To address the specificity of DYRK1A in the phosphorylation of TOM70 Ser 91 the endogenous DYRK1A should be targeted using two different CRISPR methods: first standard CRISPR KO of DYRK1A and second inducible CRISPRi targeting the promoter for DYRK1A. This should be used to analyze function for in vivo phosphorylation of endogenous TOM70 Ser 91 and TOM70 interaction with the core TOM70 import complex for mitochondrial import of receptor bound precursors.

3. The kinase specificity of INDY or DYRK1A/B must be experimentally demonstrated. Given the expertise of this group the best assay would be to use cell or tissue lysates and use Kinobeads to capture endogenous kinases discovered in the screen that have significant activity in Fig.1A (DYRK1A/B, CLK3, LIMK, RPS6KA1, CK1-delta). Multiple cell lines or tissues may need to be tested to capture the different kinases. INDY should be added at 3-5 different doses to generate a dose curve for inhibiting kinases binding to the Kinobeads similar what Kuster and co-workers did to characterize selectivity of FDA approved drugs (Science, 2017 PMID: 29191878). This experiment is required to show the specificity of INDY in the cell inhibition experiments.

4. In the discussion the authors make the bold statement that DYRK1A probably is not regulated by signaling cascades but probably expression changes and autophosphorylation of a tyrosine. Several references are cited. There is no evidence of this and must assume dimerization-like mechanism for autophosphorylation on a tyrosine residue? Either additional evidence needs to be provided for this mechanism, or the authors need to backtrack and qualify their statement. It is unclear why no signaling mechanisms could not be involved in protein interactions regulating DYRK1 activity in addition to control of expression?

As you will see from the reports copied below, the reviewers are positive but raise important concerns. We find that these concerns limit the strength of the study, and therefore we ask you to address them with additional work. Without substantial revisions, we will be unlikely to send the paper back to review. In particular, we ask that you further examine the regulation of TOM70 by DYRK1A, taking into account other possible kinases and substrates as well as performing experiments using DYRK1A knockouts and with multiple INDY concentrations, as indicated by the reviewers. In addition, we ask that you examine the role of TOM70 serine 91 phosphorylation in mitochondrial morphology and function in vivo.

If you feel that you are able to comprehensively address the reviewers' concerns, please provide a point-by-point response to these comments along with your revision. Please show all changes in the manuscript text file with track changes or colour highlighting. If you are unable to address specific reviewer requests or find any points invalid, please explain why in the point-by-point response.

> Thank you very much for the positive feedback on our manuscript. We are very happy to now submit a revised version in which we could clarify all points raised by the three reviewers. As you can see below we provide numerous novel experiments all supporting our findings, including the role of other possible kinases, analysis of DYRK1A knock out cells, titration of INDY concentrations and most importantly we provide data from several experimental approaches showing the specific role of TOM70Ser91 phosphorylation in the regulation of the mitochondrial carrier import pathway.

Reviewer #1 (Remarks to the Author):

Mitochondrial surface receptors serve as the interface between the cytosolic and the mitochondrial proteomes. They recognize cytosolic precursor proteins and, in the case of TOM70, recruit chaperones to the mitochondrial surface. TOM70 is a complicated protein with functionally distinct domains (Clamp, Core and C-terminal domain) which binds to a number of different partners, including precursors, chaperones, J-proteins, the endoplasmic reticulum, components of the ubiquitin-proteasome-system as well as Msp1/ATAD1. How these different partners are bound and how the different activities of TOM70 are coordinated is largely unknown. In this manuscript, Walter and coauthors describe a post-translational modification of human TOM70 that is of crucial relevance for its functionality. Phosphorylation of a serine residue at position 91 strongly increases its activity in the import reaction of the model substrate TIM23. It might also increase the fraction of TOM70 that forms a complex with the TOM translocase. Phosphorylation of serine 91 of TOM70 is mediated by DYRK1A/B proteases. At steady state, TOM70 appears to be predominantly in the active, i.e. phosphorylated state.

This is an exciting study which sheds light on the regulation of an important factor of the mitochondrial import machinery. Given its different functions in the context of mitochondrial protein biogenesis, TOM70 is a highly interesting candidate as regulatory control element. The identification of DYRK1A/B and the relevance of the phosphorylation of serine 91 for mitochondrial biogenesis are very convincingly shown in this study. The molecular details of this regulatory step will have to be further analyzed in the future. However, already this initial discovery study here is of very high quality and will be a significant advance for the field. Some minor points still should be considered.

> We thank the reviewer for these very positive comments. As you can see below, we aimed to clarify all points raised by the reviewer including several novel experimental data.

Specific points

1. The authors introduce INDY as specific inhibitor of DYRK1A/B activity. If their conclusion is correct than INDY-treated cells (or mitochondria) should have reduced levels of TOM70-dependent substrates. It should be straight forward to test this by Western blotting or MS.

> We agree with the reviewer that INDY treatment should lead to changes of TOM70 dependent substrates. While approaching this question of the reviewer experimentally we made the exciting observation that INDY treatment leads to a protective cellular response that includes the upregulation of all three TOM import receptors as well as several members of the DYRK kinase family acting as 'back-up' upon impaired DYRK1A signalling. We studied this unexpected and fascinating finding now in detail and include several new experiments in the revised manuscript:

a) INDY treatment does not only change the ratio from phosphorylated to non-phosphorylated TOM70 on PhosTag gels but also leads to increased TOM70 protein levels in total (Extended Data Figure 2b). Also the protein levels of the TOM20 and TOM22 import receptors increased upon INDY treatment, while the import pore TOM40 was not affected (New Figure 3b). For TOM70 and TOM20 we observe also increased transcript levels pointing to a protective cellular response (New Figure 3c). Consequently the import of the carrier model substrate TIM23 is only mildly affected (New Figure 3a).

Extended Data Figure 2b. PhosTag gel analysis of TOM70 protein levels upon INDY treatment indicates a protective response resulting in overexpression of TOM70.

New Figure 3b. Western blot analysis of isolated mitochondria (Mito.) and whole cell extracts (WCE) after INDY treatment (24 h) reveals increased protein levels of TOM70, TOM22, TOM20 and the DYRK1A isoform DYRK1B.

New Figure 3c. qPCR analysis after INDY treatment reveals upregulation of the two peripheral import receptors TOM70 and TOM20 already on transcript level.

New Figure 3a. Import of radiolabeled TIM23 into mitochondria isolated from cells after INDY treatment is only mildly affected.

We then aimed to treat cells with INDY while simultaneously inhibiting the expression of *TOMM70* via specific shRNAs. In this setup, we could indeed observe less phosphorylated TOM70 and consequently impaired import of TIM23 (New Figures 3d and e; see Response Reviewer 2, point 3).

In addition we tested endogenous levels of TIM23, our model substrate of the TOM70-dependent carrier import pathway. After prolonged incubation of cells with INDY (72 h) we could indeed find mildly reduced level of TIM23 by western blot analysis (new Figure R1). A stronger reduction of carrier proteins can not be expected due to the identified cellular response upon INDY treatment.

New Figure R1. Western blot analysis of mitochondria isolated from cells grown in the presence or absence of INDY (72 h) shows decrease of the carrier substrate TIM23 but not SDHA or TOM40.

2. Does the overexpression of TOM70 suppress the INDY-induced mitochondrial morphology phenotype? This would be expected if the DYRK1A/B role is indeed exclusively explained by its activation of TOM70.

> We thank the reviewer for this suggestion. We have performed this experiment and find that overexpression of TOM70^{WT} but not the TOM70^{Ser91Ala} variant can largely suppress the INDY-induced phenotype (New Figures 2b,c). Notably, overexpression of both TOM70 constructs per se did not affect mitochondrial morphology (New Figure 2b and New Extended Data Figure 2d).

For further experimental in vivo data supporting a specific role of the Ser91 residue of TOM70 in protein import we would like to refer here to the response to reviewer 2, point 4.

New Figures 2b (top panel) and c (middle panel) and new Extended Data Figure 2d (lower panel). In vivo fluorescence microscopy analysis of mitochondrial morphology of U2OS cells after INDY treatment and in combination with overexpression of TOM70^{WT} or TOM70^{Ser91Ala}. While the mitochondrial tubular network is strongly affected by INDY treatment, overexpression of TOM70^{WT} partially prevents these effects. In contrast, cells overexpressing the Ser91Ala TOM70 mutant are not protected and display the same morphology as the empty vector (e.v.) control.

3. The authors nicely show that phosphorylation is not relevant for the ability of TOM70 to bind substrates (in vitro). However, they did not test any relevance for the chaperone-binding Clamp domain which is in direct proximity to the serine 91 residue. However, a (presumably the) critical function of Tom70 is the binding of chaperones via its EEVD-binding Clamp domain. To analyze the effect of serine 91 phosphorylation on chaperone binding is beyond the scope of this initial study, but it should be discussed.

> We thank the reviewer for this positive statement. We have stressed this point in the discussion part and emphasize the critical role of TOM70 in chaperone binding.

4. The data on the TOM70 phosphorylation by DYRK1A/B are very sound and convincing. However, the evidence of a phosphorylation-induced TOM70 recruitment to TOM22 is not. The latter is just based on a somewhat stronger band in the pull-down shown in Fig. 3g. The increase in the pull-down signals in Fig. 3h is strong from the quantification, but actually, the signals of the band in the +LP sample looks stronger to me than those after the DYRK1A treatment. Since the authors have the INDY inhibitor, it should be straight forward to improve the evidence for the DYRK1A/B-dependent TOM70 recruitment.

> We provide now further experimental data showing phosphorylation-dependent recruitment of TOM70 to the TOM core complex:

- We tested the pull-down experiment in HEK293T mitochondria and find also here increased binding of phosphorylated TOM70 to the TOM complex confirming our findings from mouse brain mitochondria (see New Extended Data Figures 4 e and f, response to reviewer 2, point 5). The combination of in vivo INDY treatment and pull-down experiments suggested by the reviewer is not applicable as INDY treatment leads to upregulation of TOM70 and further TOM subunits as described above.

- Based on the subsequent comment (point 5) of this reviewer, we followed the strategy designed by Wiedemann et al. (EMBO J. 2001) that uses the possibility to arrest a carrier precursor protein at the import machinery and directly monitor transfer from the Tom70 dimer to the import pore. This is mediated by blocking the unfolding of a DHFR domain that is fused to the precursor (by addition of methotrexate). The precise position of the precursor arrest can be monitored by Blue Native PAGE (New Figure 5e). With this approach we can now clearly show the role of TOM70 phosphorylation

for the transfer of precursor from the TOM70 dimer to the TOM complex in addition to the two different Co-IP approaches.

New Figure 5e. Arrest of the ATP/ADP carrier 2 (Aac2) precursor protein fused to DHFR domain (Wiedemann et al., 2001) at the TOM complex and TOM70 dimer in the presence of methotrexate (MTX). Treatment of Lambda Phosphatase (LP) prior to the import reaction leads to decreased TOM arrest but increased arrest of precursor at the TOM70 dimers (compare lane 5 to lane 1) confirming our model that phosphorylation is required for the precursor transfer from the receptor bound state to the TOM complex. *, Star indicates assembled Aac2-DHFR in the inner membrane (Prot. K protected, lanes 2 and 4) in the control (Ctrl.) condition. Prot. K, Proteinase K sensitivity serves as control for exposure of the arrested precursor at the outer membrane (compare lanes 1 and 5 to lanes 3 and 7, respectively).

5. In the past, crosslinking was very powerful to show the binding of carriers to TOM70 (see for example Fig. 1A of Wiedemann et al, EMBO J 20, 951f). This simple assay could be a powerful basis to provide convincing evidence for the claim that phosphorylation of TOM70 is not required for carrier binding, however crucial for the subsequent TOM70 -> TOM40 transfer. This should be considered.

> We would like to refer to point 4 above in which we show the specific role of TOM70 phosphorylation for precursor transfer from the receptor bound state to the import pore according to the experimental set up designed by Wiedemann et al. (2001).

Minor points:

6. The microscopy image in Fig. 2b is too dark. The mitochondria are hardly visible, at least on my screen

> Based also on point 2 of this reviewer we provide now an entire new data set for the microscopy experiments.

7. Page 5, five lines from bottom: For consistency, Tom70S110A should be TOM70S110A

> We thank the reviewer for careful reading. We have corrected this inconsistency in the revised version of the manuscript.

8. Page 6, top line: gelelectrophoresis should be gel electrophoresis

> We have corrected this in the revised version.

Reviewer #2 (Remarks to the Author):

Walter et al. performed in vitro profiling of the human kinome and identified that DYRK1A is able to phosphorylate TOM70 with high efficiency. Through combination of LC-MS for phosphorylation site identification and DYRK1A treatment of TOM70 bearing different mutation site, the authors showed that DYRK1A-dependent phosphorylation of TOM70 happens specifically at Ser91. Further experiments showed that Ser91 phosphorylation of TOM70 would increase its binding to the TOM complex. Finally, a simple model about DYRK1A-dependent carrier import pathway was proposed according to these observations.

The authors made an interesting discovery. However, mechanistic details upstream and downstream of this phosphorylation event is necessary to make the story complete. Also, more evidence is needed to support the assumptions made by the authors. I suggest major revision should be made before it meets the publication criteria of Nature Communications.

> We thank the reviewer for appreciating our finding as an interesting discovery. We also would like to thank for the very constructive comments. As you can see, we included now many novel experimental data to analyze the role of TOM70 phosphorylation at a molecular level upstream and downstream of this event.

1. It's not quite accurate to say that "Human TOM70 is nearly quantitatively phosphorylated at Ser91.". As indicated by LC-MS results and the existence of bands

containing multiple phosphor groups during phos-tag gel-electrophoresis of untreated mitochondria (Figure 1d), it seems that, while nearly all TOM70 is phosphorylated, Thr85 and Ser110 also contribute to the phosphorylation of TOM70. The band in the Ser91-specific antibody staining cannot rule out the possibility that mono-phosphorylated TOM70 at Thr85 or Ser110 exists. Further evidence is necessary to support the statement that TOM70 is nearly quantitatively phosphorylated at Ser91. Authors may try to use MS-based quantitative method like MRM-MS to explain this.

> It is of course possible, as the reviewer points out, that mono phosphorylated TOM70 at Thr85 or Ser110 exists and we do not claim in our manuscript that this is not the case. We focus here on the Ser91 site, which is nearly quantitatively phosphorylated by DYRK1A. However, we have realized that the arrangement of the TOM70 and TOM70^{pSer91} western blots do not simply allow comparison of the size of the bands. We are sorry if this was misleading. We provide now the New Extended Data Figure 1f in which we loaded the samples and arranged the blots in a way that allows a direct comparison of the migration of phosphorylated and non-phosphorylated TOM70 in mitochondria. With this, one can now clearly see that TOM70 is (almost) quantitatively phosphorylated at position Ser91 and that the respective signal is lost after LP treatment when probing with the TOM70^{pSer91} specific antibody.

New Extended Data Figure 1f. Western blot analysis of PhosTag gel of mitochondria incubated in the presence or absence of lambda phosphatase (LP). Immunodecoration with TOM70 and TOM70^{pSer91} antibodies reveals almost quantitative phosphorylation of TOM70^{Ser91}.

2. Overall the authors haven't adequately addressed the issue of other kinases that might phosphorylate TOM70 and other substrates DYRK1A can phosphorylate that can interference with functional assays of TOM70 phosphorylation. For example, while DYRK1A/B received the highest score in the screening assay, HIPK1 is a close second. HIPK1 also belongs to the DYRK family. Have the authors look into HIPK1 as well for potential cross reactivity? As indicated by the authors, DYRK1A is implicated in several diseases and there are multiple reported DYRK1A substrates including MAP1B. The functional assays through inhibition of DYRK1A (e.g., Fig. 2d and Extended Data Fig.2a) do not necessarily attribute to the change of TM70 phosphorylation.

> We thank the reviewer for raising this issue. We provide now new experimental data that show that

a) HIPK1 and further DYRK family kinases which scored below DYRK1A/B in our *KinaseFinder* assay can also phosphorylate TOM70 Ser91, however, to a lesser extent as DYRK1A/B (New Extended Data Figure 1h) and

b) inhibition of DYRK1A by INDY treatment induces not only upregulation of the three import receptors TOM70, TOM20 and TOM22 but also of other DYRK family kinases, including DYRK1B, DYRK2, HIPK1 and the related CLK2 (New Figures 3b,c (see comment to reviewer 1 above) and new Extended Data Figure 3). This led to the exciting discovery that inhibition of DYRK1A causes a protective cellular response that

includes remodeling of the mitochondrial import machinery and the activation of 'back-up' kinases of the DYRK family, which can counterbalance loss of TOM70 Ser91 phosphorylation. These results highlight the importance of TOM70 Ser91 phosphorylation as a critical activator of the carrier import pathway in human mitochondria.

New Extended Data Figure 1h. Analysis of mitochondria that have been treated with lambda phosphatase (LP) followed by incubation with the indicated kinases. PhosTag gels and TOM70^{pSer91} specific antibodies reveal potency of DYRK family kinase HIPK1 and CLK2 to phosphorylate this position while the various PKC isoforms did not show detectable activity.

New Extended Data Figure 3. qPCR analysis showing increased expression of transcripts of DYRK family kinases upon INDY treatment.

Regarding other, non-mitochondrial substrates of DYRK1A we have profiled the known substrates p53 and Tau (Ogawa et al., 2010; Park et al., 2010) (the suggested MAP1B was too large to analyze on PhosTag gels). DYRK1A inhibition by INDY did not change migration of both tested substrates on PhosTag gels (New Figure R2). We also tested for changes in transcript levels of known DYRK1A substrates and found a slight reduction for Tau but no changes for the other substrates tested (New Figure R3).

New Figure R2. PhosTag analysis of annotated DYRK1A substrates Tau and p53 in cells that have been incubated in the presence or absence of INDY.

New Figure R3. qPCR analysis of transcripts of annotated DYRK1A substrates.

Regarding the issue to attribute TOM70 Ser91 phosphorylation to functional assays we would like to refer to the new microscopy data, showing that overexpression of TOM70^{WT} but not TOM70^{Ser91Ala} can suppress the changes in mitochondrial structure upon INDY treatment (New Figure 2b,c; New Extended Data Figure 2d, see above Reviewer 1) and its direct involvement in the carrier import pathway (New Figure 5g, see below). In summary, our new results confirm TOM70 Ser91 as an important target of DYRK1A playing an essential role to activate the carrier import pathway in human mitochondria.

3. In Figure 2a, Ser91 phosphorylated TOM70 still exists after INDY treatment. Is it due to insufficient inhibition of DYRK1A or that there is other kinase(s) responsible for Ser91 phosphorylation of TOM70? Did the authors try to raise the concentration of INDY or perform a DYRK1A-knockout experiment?

> As pointed out above (point 2 and see also Reviewer 1, point 1) INDY treatment causes a protective response that includes upregulation of TOM70 transcripts but also of other TOM receptors as well as upregulation of further DYRK family kinases which also phosphorylate TOM70 Ser91 as 'back-up' kinases. Increasing INDY concentrations was therefore not effective. We have performed the same INDY treatment in the presence of TOM70 shRNA to suppress the INDY-mediated transcriptional response to raise endogenous TOM70. See New Fig. R4 showing efficiency of *TOMM70* gene silencing. With this we succeeded to achieve an INDY-specific reduction of the level of phosphorylated TOM70 (New Figure 3d) and consequently impairment of the carrier import pathway (New Figure 3e).

New Figure R4. Control western blots of HEK293T cells showing efficient knockdown of TOM70 expression by *TOMM70* shRNA. As control (Ctrl.) non-silencing shRNA was applied. Gene silencing was induced by incubation with doxycycline (1 µg/ml) for indicated time.

New Figure 3d. TOM70 Phos-tag gels (lanes 1 and 2) and SDS-PAGE analysis (lanes 3 and 4) of mitochondria isolated from cells grown overnight in the presence or absence of INDY (10 μ M) and, for both conditions, in the presence of TOM70 shRNA.

New Figure 3e. Import of TIM23 is impaired in mitochondria isolated from TOM70 shRNA-induced cells when incubated in the presence of INDY.

4. Direct evidence, like in vivo TOM70 S91A mutant experiments, connecting loss of Ser 91 phosphorylation of TOM70, and impairment of mitochondria structure and functions is necessary. DYRK1A have many substrates and inhibition of it would lead to down-regulation of phosphorylation of many proteins.

> We refer here to our response and novel experimental data provided in point 2 and 3 above as well as to reviewer 1, point 1. Moreover, we show now new experimental data that expression of the TOM70^{Ser91Ala} variant in TOM70 knock-down cells leads to impairment of the carrier import pathway compared to expression of TOM70^{WT}. The presequence import pathway, that does not require TOM70 as import receptor, is not affected (see new Figure 5g and New Extended Data Figure 5). This in vivo approach fully confirms the activatory role of TOM70 Ser91 in the carrier import pathway.

New Figure 5g. TIM23 import is impaired in mitochondria from TOM70 knock-down cells (shRNA mediated) re-expressing the Ser91Ala variant compared to the wild-type (WT).

New Extended Data Figure 5. Presequence import pathway tested by Hsp10 is not affected in mitochondria from TOM70 knock-down cells re-expressing the Ser91Ala variant compared to the wild-type (WT).

5. Assessment of TOM70 binding with TOM complex was performed using mouse brain mitochondria is not reasonable. Using mitochondria from HEK293T cells is necessary. Data from different species shouldn't be simply combined.

> We include now new pulldown experiments showing that also in mitochondria from HEK293T cells TOM70 binding to the TOM complex depends on phosphorylation as requested by the reviewer (New Extended Data Figures 4e,f). Moreover, we refer to our new experimental approach using carrier precursor-DHFR fusions which allows to study on a molecular level the transition of carrier precursor from the receptor bound state to the translocation pore. And exactly this step was found to depend on phosphorylation (see response Reviewer 1, point 4; New Figure 5e).

New Extended Data Figure 4f.

Immunoprecipitation of TOM complex via TOM22 antibodies from HEK293T mitochondria that were incubated before in the presence of LP (TOM70 dephosphorylated) or DYRK1A (TOM70 phosphorylated). Samples were analyzed by SDS-PAGE followed by immunoblotting with indicated antibodies. L, load; FT, flowthrough; W, wash; E, elution.

New Extended Data Figure 4e. Phos-tag analysis of TOM70 from samples used for immunoprecipitation from HEK293T mitochondria

6. It seems that the amount of imported TIM23 increased significantly between 20 and 45 mins without DYRK1A treatment, as shown in Figure 4e, which is contradict to what shown in Figure 4b and 4c. What's the possible reasons?

> As gold standard in the field, mitochondrial protein import should be measured by kinetics analyses, which we did in both experiments referred to by the reviewer (Fig. 4e (now new Fig. 4f) and Fig. 4b). We therefore see increasing signal intensities of the imported radiolabelled precursor over time. However, when an experimental manipulation, like phosphatase (Fig. 4b) or kinase treatments (Fig. 4f) lead to strong changes in the import efficiency that under one condition (e.g. phosphatase treatment) the signal is hardly visible anymore or under another condition (e.g. kinase treatment) the import is heavily stimulated we need to adapt the exposure times during autoradiography to keep the import kinetic in a linear range (as far as this is possible when such dramatic changes like here are observed).

7. While lambda phosphatase treatment is effective in removing phosphorylation, it is neither substrate-specific nor site-specific. Direct evidence suggesting Ser91 phosphorylation is responsible for TOM70 binding of TOM complex and TIM23 import should be provided, since the dephosphorylation and subsequent phosphorylation of other protein(s) within the mitochondria could be (one of) the cause(s). Besides, was the level of TIM23 import after DYRK1A treatment comparable with that before lambda phosphatase treatment?

> We agree with the reviewer that lambda phosphatase, like most phosphatases, is a relatively unspecific enzyme. Most important for our experimental design is that DYRK1A has no further target at the mitochondrial import machinery except TOM70 Ser91 (shown by several different approaches including MS profiling of DYRK1A dependent substrates in the presence of γ -[$^{18}\text{O}_4$]-ATP (see Figures 1 e,f) and that the two other TOM receptors seem not to be phosphorylated in mitochondria (Extended Data Figure 1e)). Therefore, the reconstitution of the carrier import pathway by DYRK1A after removal of the quantitatively phosphorylated TOM70 Ser91 position by lambda phosphatase can indeed allow conclusions related to the mitochondrial import machinery. In addition, we provide now novel experimental data showing direct evidence for the specific role of TOM70 Ser91 in protein import as well as mitochondrial morphology (New Figure 5g and New Figure 2 b,c) fully confirming our findings. When testing import activities under different conditions (e.g. +/- phosphatase treatment) the same amount of mitochondria were used to have comparable number of import sites, including TIM23. In Fig. 5f, which we assume the reviewer refers to, all samples (equal amount of mitochondria and thus also import sites) were first treated with phosphatase and then further incubated in the absence (-) or presence (+) of DYRK1A. The level of endogenous TIM23 is therefore not changed. The shown TIM23 signal reflects the increased import rates of radiolabelled TIM23 precursor, which is added to the reaction externally to monitor its import and assembly rates. We hope this clarifies the question of the reviewer.

8. Whether DYRK1A is the only kinase of TOM70 is not clearly discussed. This criticism is due to the lack of appropriate controls in some of the new key experiments presented. It is necessary to knockout DYRK1A and show its affection on the phosphorylation of TOM70.

> Regarding the first point we would like to refer the reviewer to his/her point 2 (above) where we answered this issue in detail including several novel experimental data figures.

Regarding the second point we generated a *DYRK1A*^{-/-} knock-out cell line using CRISPR-Cas9 and can confirm all our central findings including reduced level of TOM70 Ser91 phosphorylation, impairment of the carrier import pathway as well as a partial increase of the transcript level of further DYRK family kinases and a severe increase of DYRK1B protein level (see response to reviewer 3, point 2).

9. In the main text (P5), the author said the phosphopeptide is enriched by ERLIC but in the method part, the phosphopeptide is enriched by TiO2.

> We actually used both procedures: We used the enrichment of phosphopeptides by ERLIC for the identification of phospho sites in vivo and describe this procedure in the method part in the section *Identification of TOM70 phosphorylation sites in HEK293T*

cells. The enrichment by TiO₂ was used for the *In vitro* γ -[¹⁸O₄]-ATP kinase assay and described under this headline.

Reviewer #3 (Remarks to the Author):

This is a well done study characterizing the phosphorylation of Ser91 of TOM70 by DYRK1A and showing this regulates TOM70 interaction with the core TOM70 import complex for mitochondrial import of receptor bound precursors. The functional role for DYRK1A phosphorylation and regulation of TOM70 is potentially an important discovery for dysfunctional DYRK1A including DYRK1A-related syndrome, Autism Spectrum Disorder and Down syndrome.

> We thank the reviewer for this positive evaluation.

Specific recommendations to improve the manuscript:

1. The kinase screen in Fig.1A identifies DYRK1A/B as having the highest activity ratios in the screen. But several other kinases have very significant activity including HIPK1, PKC-beta, other PKCs and CLK. But in Fig. 1B several kinases are compared to DYRK1A/B for comparison TOM70cd Ser 91 that have very low activity ratios in the screen (CLK3, LIMK, RPS6KA1, CK1-delta). The phosphorylation assay in Fig.1B should be shown for HIPK1, the different PKCs and CLK. It is possible that additional kinases have the ability to phosphorylate TOM70 Ser 91 and regulate functional TOM70 activity. This must be tested.

> We have included the requested experiment (New Extended Data Figure 1h) showing that HIPK1, a member of the DYRK kinase family, and CLK2, which belongs to a related kinase family, can in principle also phosphorylate TOM70 Ser91, however, to a lesser extent. All PKC isoforms tested did not display any activity in phosphorylating TOM70 Ser91. In addition we performed the *KinaseFinder* assay for the purified receptor domain of a TOM70^{Ser91Ala} variant and find that all PKC isoforms were still highly active, while DYRK1A and DYRK1B as well as further members of the DYRK family kinases and CLK2 relocated in the background noise (New Extended Data Figure 1g). Thus, PKC seems to target the purified TOM70 receptor domain at another, still unknown position. For PKC we also performed an *in vivo* inhibition experiment using the well established PKC inhibitor Gouml6983 and find that TOM70 Ser91 phosphorylation is not affected when PKC is inhibited (New Figure R5). Most excitingly, and outlined in detail above, we find that specific inhibition of DYRK1A not only causes remodeling of the mitochondrial import machinery but also leads to an upregulation of DYRK1B and exactly these additional members of the DYRK family and the related CLK2, which were found in the *KinaseFinder* screen.

New Extended Data Figure 1g. Results of *KinaseFinder* assay (ProQinase™) for the Ser91Ala variant of human TOM70_{cd}. Shown are activity ratios of ³³P-ATP-radiometric filter binding assays for 339 different protein kinases tested. Dashed line indicates threshold for relevant activities.

New Figure R5. Presence of the PKC inhibitor *Gouml 6983* (Gafni et al., 2013) in cell culture does not affect TOM70^{Ser91} phosphorylation (PhosTag gel, upper panel). Efficient decrease of phosphorylated PKC substrates in the presence of *Gouml 6983* was controlled with a specific Phospho-Ser-PKC antibody (lower panel).

In summary, our new experimental data reveal the importance of the DYRK1A-mediated phosphorylation of TOM70, which is secured by a protective cellular response upon inhibition of DYRK1A at the level of (i) upregulation of TOM70 and further import receptors like TOM20 and TOM22 to increase the number of potential precursor binding sites at mitochondria and (ii) upregulation of *back-up* kinases which are also capable to phosphorylate TOM70 Ser91 (and exactly these ones were found as additional (medium score) hits below DYRK1A/B in our *KinaseFinder* screen (Fig. 1a)).

2. To address the specificity of DYRK1A in the phosphorylation of TOM70 Ser 91 the endogenous DYRK1A should be targeted using two different CRISPR methods: first standard CRISPR KO of DYRK1A and second inducible CRISPRi targeting the promoter for DYRK1A. This should be used to analyze function for *in vivo* phosphorylation of endogenous TOM70 Ser 91 and TOM70 interaction with the core

TOM70 import complex for mitochondrial import of receptor bound precursors.

> As outlined above, we find that inhibition of *DYRK1A* leads, beside upregulation of *TOMM70* transcripts, also to an upregulation of *DYRK1B* and further DYRK family kinases (New Figures 3b,c; New Extended Data Figure 3) so that the level of TOM70 Ser91 phosphorylation is maintained (Extended Data Figure 2b). A specific change in Ser91 phosphorylation can only be observed if *TOMM70* expression is simultaneously impaired by shRNA (New Figure 3d) which leads to impairment of the carrier import pathway (New Figure 3e) fully confirming our findings.

In addition, we provide new experimental data for the specific role of the TOM70 Ser91 position in mitochondrial protein import (New Fig. 5g) as well as mitochondrial morphology (Figures 2b,c). We also generated, as requested by the reviewer, a *DYRK1A*^{-/-} cell line using CRISPR-Cas9. We find a severe reduction of TOM70 Ser91 phosphorylation and consequently impairment of the carrier import pathway (New Figures R6 and R7). However, also here we find an upregulation of *DYRK1B* and other 'back-up' kinases (New Figure R8), similarly to the chemical inhibition by INDY (as described above) to secure the TOM70 Ser91 phosphorylation. Thus, also the *DYRK1A* KO approach fully confirms our findings.

New Figure R6. Generation and characterization of a *DYRK1A*^{-/-} cell line. **Top panel**, strategy to target human *DYRK1A* by CRISPR-Cas9. Guide RNA was directed against exon 6 of *DYRK1A*. Successful targeting of the *DYRK1A* gene was confirmed by sequencing. **Bottom panel**, western blot analysis of HEK293T cells after SDS-PAGE reveals absence of immunoreactive *DYRK1A* signal and increased level of *DYRK1B*. PhosTag analysis revealed decrease of TOM70^{Ser91} phosphorylation.

New Figure R7. Impaired TIM23 import in mitochondria isolated from *DYRK1A*^{-/-} cells.

New Figure R8. qPCR analysis of transcripts from DYRK family members and CLK2 in *DYRK1A*^{-/-} cells.

3. The kinase specificity of INDY or DYRK1A/B must be experimentally demonstrated. Given the expertise of this group the best assay would be to use cell or tissue lysates and use Kinobeads to capture endogenous kinases discovered in the screen that have significant activity in Fig.1A (DYRK1A/B, CLK3, LIMK, RPS6KA1, CK1-delta). Multiple cell lines or tissues may need to be tested to capture the different kinases. INDY should be added at 3-5 different doses to generate a dose curve for inhibiting kinases binding to the Kinobeads similar what Kuster and co-workers did to characterize selectivity of FDA approved drugs (Science, 2017 PMID: 29191878). This experiment is required to show the specificity of INDY in the cell inhibition experiments.

> We now show in the New Extended Data Figure 2a a titration experiment of different INDY concentrations applied directly to the phosphorylation assay in the presence of recombinant DYRK1A and the purified TOM70 receptor domain. The result clearly shows a concentration dependent inhibition of DYRK1A activity by INDY fully confirming our results.

New Extended Data Figure 2a. Titration of indicated INDY concentrations to analyze DYRK1A activity to phosphorylate human TOM70 receptor domain.

As further control we tested if INDY might also inhibit non-related kinases and find that it does not affect the activity of casein kinase II (CKII; new Figure R9). Also for the analysis of further kinases throughout the entire manuscript we used recombinant kinases and profiled their activities directly with purified receptor domains or isolated mitochondria and tested their specificity with phosphospecific antisera.

New Figure R9. Control experiment for new Extended Data Figure 2a showing that INDY at max. concentration (10 μ M) can not inhibit Casein Kinase 2 (CK2). Yeast Tom22 receptor domain served here as specific CK2 target (Schmidt et al., 2011).

4. In the discussion the authors make the bold statement that DYRK1A probably is not regulated by signaling cascades but probably expression changes and autophosphorylation of a tyrosine. Several references are cited. There is no evidence of this and must assume dimerization-like mechanism for autophosphorylation on a tyrosine residue? Either additional evidence needs to be provided for this mechanism, or the authors need to backtrack and qualify their statement. It is unclear why no signaling mechanisms could not be involved in protein interactions regulating DYRK1 activity in addition to control of expression?

> We agree that the literature regarding this issue is quite limited and therefore omitted this part on the reviewer's advice.

References

- Gafni, O. et al. Derivation of novel human ground state naïve pluripotent stem cells. *Nature* **504**, 282-286 (2013).
- Ogawa, Y. et al. Development of a novel selective inhibitor of the Down syndrome-related kinase Dyrk1A. *Nat. Commun.* **1**, 86 (2010).
- Park, J., Oh, Y., Jung, M.S., Song, W.J., Lee, S.H., Seo, H & Chung, K.C. Dyrk1A Phosphorylates p53 and Inhibits Proliferation of Embryonic Neuronal Cells *J. Biol. Chem.* **285**, 31895-31906 (2010).
- Schmidt, O., Harbauer, A.B., Rao, S., Eyrich, B., Zahedi, R.P., Stojanovski, D., Schönfisch, B., Guo, B., Sickmann, A., Pfanner, N. & Meisinger, C. Regulation of mitochondrial protein import by cytosolic kinases. *Cell* **144**, 227-239 (2011).
- Wiedemann, N., Pfanner, N. & Ryan, M.T. The three modules of ADP/ATP carrier cooperate in receptor recruitment and translocation into mitochondria. *EMBO J.* **20**, 951–960 (2001).

REVIEWERS' COMMENTS

Reviewer #1 (Remarks to the Author):

The authors addressed all issues raised on the previous submission. This is a very exciting study of excellent quality. I support publication in its present form.

Reviewer #2 (Remarks to the Author):

The authors have properly addressed all comments from this reviewer and provided many additional experimental evidence to make this work complete. I would like to suggest its publication.

REVIEWERS' COMMENTS

Reviewer #1 (Remarks to the Author):

The authors addressed all issues raised on the previous submission. This is a very exciting study of excellent quality. I support publication in its present form.

Reviewer #2 (Remarks to the Author):

The authors have properly addressed all comments from this reviewer and provided many additional experimental evidence to make this work complete. I would like to suggest its publication.

> we thank the reviewers for their very positive comments and suggestions that helped to improve our manuscript